# Learning to Adapt: In-Context Learning Beyond Stationarity

**Zhen Qin**[†,‡,*]**, Jiachen Jiang**[‡]**, Zhihui Zhu**[‡]

[†]Michigan Institute for Computational Discovery and Engineering,
Department of Electrical Engineering and Computer Science,
Department of Statistics, University of Michigan
[‡]Department of Computer Science and Engineering, The Ohio State University

## Abstract

Transformer models have become foundational across a wide range of scientific and engineering domains due to their strong empirical performance. A key capability underlying their success is in-context learning (ICL): when presented with a short prompt from an unseen task, transformers can perform per-token and next-token predictions without any parameter updates. Recent theoretical efforts have begun to uncover the mechanisms behind this phenomenon, particularly in supervised regression settings. However, these analyses predominantly assume stationary task distributions, which overlook a broad class of real-world scenarios where the target function varies over time. In this work, we bridge this gap by providing a theoretical analysis of ICL under non-stationary regression problems. We study how the gated linear attention (GLA) mechanism adapts to evolving input-output relationships and rigorously characterize its advantages over standard linear attention in this dynamic setting. To model non-stationarity, we adopt a first-order autoregressive process and show that GLA achieves lower training and testing errors by adaptively modulating the influence of past inputs—effectively implementing a learnable recency bias. Our theoretical findings are further supported by empirical results, which validate the benefits of gating mechanisms in non-stationary ICL tasks.

## 1 Introduction

Transformer-based architectures (Vaswani et al., 2017) have emerged as a powerful and versatile modeling framework, achieving state-of-the-art results across a wide spectrum of scientific and engineering domains. Their remarkable effectiveness has been demonstrated in natural language processing (Radford et al., 2019; Brown et al., 2020), recommendation systems (Zhou et al., 2018; Chen et al., 2019), reinforcement learning (Chen et al., 2021; Janner et al., 2021), computer vision (Dosovitskiy et al., 2020), and multi-modal signal processing (Tsai et al., 2019), as well as in more specialized areas such as quantum information (Ma et al., 2025) and wireless communication systems (Kim et al., 2023). A particularly notable instance is their pivotal role in the development of large language models like GPT-4 (Achiam et al., 2023), where the Transformer backbone enables highly advanced generative capabilities.

A distinctive and increasingly studied feature of Transformer models is in-context learning (ICL) (Min et al., 2021), which allows the model to perform previously unseen tasks at inference time by conditioning on sequences of input-output examples, without requiring any explicit parameter updates. This emergent capability has spurred a growing body of research aiming to understand the underlying mechanisms that enable such behavior (Brown et al., 2020; Min et al., 2021; Dong et al., 2022; Wies et al., 2023; Zhang et al., 2023; Bai et al., 2023; Li et al., 2024a; Bertsch et al., 2024; Akyürek et al., 2024; Jiang et al., 2025a; Song et al., 2024; Wu et al., 2023; Qin et al., 2025). In particular, recent theoretical works have investigated the realization of ICL in supervised regression settings, showing that certain architectural components–such as linear attention mechanisms–can effectively emulate simple learning algorithms, e.g., a single step of gradient descent, when the input data distribution is stationary (Garg et al., 2022; Akyürek et al., 2022; Von Oswald et al.,

---

*Corresponding Author: zhenqin@umich.edu

2023; Zhang et al., 2024; Huang et al., 2023; Chen et al., 2024; Yang et al., 2024; Zhang et al., 2025; Mahankali et al., 2023; Ahn et al., 2023; Li et al., 2024b; 2025; Fu et al., 2024; Ding et al.). These findings offer valuable insights into the algorithmic behaviors implicitly encoded by architectural design, shedding light on the interplay between representation, memory, and adaptation in modern Transformer models.

However, much of the existing theoretical understanding is limited to stationary data settings, where the input-output relationships remain consistent across in-context examples and the query point. In contrast, many practical scenarios–including time-series forecasting, streaming data, and natural language–exhibit non-stationarity, where the underlying target function evolves over time. In such settings, recency bias, or the increased predictive relevance of more recent examples, plays a crucial role in accurate prediction. Empirically, linear attention mechanisms are often insufficient for these non-stationary tasks, motivating the introduction of architectural variants that incorporate inductive biases better suited for adaptation, such as gated linear attention (GLA) (Yang et al., 2023; Jiang et al., 2025b), RetNet (Sun et al., 2023), Gateloop (Katsch, 2023), RWKV-6 (Peng et al., 2024), as well as state-space models like Mamba-2 (Gu & Dao, 2023). These methods have achieved strong performance in non-stationary sequence modeling, yet there remains a lack of formal theoretical understanding of their behavior in ICL settings.

**Contribution: In this paper, we aim to bridge this gap by presenting a theoretical analysis of ICL in non-stationary or time-varying regression problems.** We investigate how the GLA mechanism adapts to evolving input-output relationships and provide a rigorous characterization of its advantages over standard linear attention in this setting. To model non-stationarity, we adopt a first-order autoregressive process, which allows us to analytically capture temporal variations in the regression targets. Within this framework, we show that standard linear attention incurs higher training and testing errors due to its limited capacity to adapt to distributional shifts over time. In contrast, GLA exhibits inherent adaptability by dynamically modulating the contributions of past inputs, effectively inducing a learnable recency bias. This gating mechanism enables the model to better accommodate time-varying input-output mappings, thereby achieving more robust in-context generalization. Our analysis underscores the importance of architectural components–particularly gating–in equipping transformer models with the ability to implement adaptive learning algorithms in non-stationary environments. Experimental results further corroborate our theoretical findings. Collectively, our work contributes a theoretical perspective that clarifies the design choices behind transformer variants and offers a conceptual framework for understanding and developing architectures suited for adaptive ICL.

**Notation** We use bold capital letters (e.g., $\boldsymbol{Y}$) to denote matrices, bold lowercase letters (e.g., $\boldsymbol{y}$) to denote vectors, and italic letters (e.g., $y$) to denote scalar quantities. Elements of matrices are denoted in parentheses, as in Matlab notation. For example, $\boldsymbol{Y}(s_1, s_2)$ denotes the element in position $(s_1, s_2)$ of the matrix $\boldsymbol{Y}$. The inner product of $\boldsymbol{A} \in \mathbb{R}^{d_1 \times d_2}$ and $\boldsymbol{B} \in \mathbb{R}^{d_1 \times d_2}$ can be denoted as $\langle \boldsymbol{A}, \boldsymbol{B} \rangle = \sum_{s_1=1}^{d_1} \sum_{s_2=1}^{d_2} \boldsymbol{A}(s_1, s_2) \boldsymbol{B}(s_1, s_2)$. $\|\boldsymbol{X}\|_F$ represents the Frobenius norm of $\boldsymbol{X}$. $\boldsymbol{0}_d$ and $\boldsymbol{0}_{d \times d}$ denote the zero vector in $\mathbb{R}^d$ and the zero matrix in $\mathbb{R}^{d \times d}$, respectively. For a positive integer $K$, $[K]$ denotes the set $\{1, \dots, K\}$.

## 1.1 RELATED WORKS

A growing body of work has investigated the emergent phenomenon of ICL, with a focus on understanding its behavior in stationary regression tasks. For example, (Garg et al., 2022) empirically demonstrated the ICL capabilities of transformers by analyzing prompts where each input is labeled by a task-specific function drawn from a predefined function class, such as linear models. Along similar lines, (Akyürek et al., 2022) investigated linear regression and introduced a transformer construction capable of performing a single gradient descent (GD) step using in-context examples. Building upon this, (Von Oswald et al., 2023) designed weight matrices for linear attention-only transformers that replicate GD updates in linear regression tasks, and notably, they observed that the learned weights resemble those obtained through end-to-end training on ICL prompts.

Further progress has been made by studying the convergence behavior of transformer architectures. In particular, (Zhang et al., 2024) showed that, for a single-layer linear self-attention model, gradient flow with carefully chosen random initialization converges to a global minimum, yielding low prediction error on anisotropic Gaussian data. Complementary work by (Huang et al., 2023) initiated the theoretical study of softmax attention, analyzing the training dynamics of one-layer, single-head

transformers and providing convergence guarantees for linear regression. This line of research was subsequently extended by (Chen et al., 2024; Yang et al., 2024; Zhang et al., 2025), who provided sufficient conditions for the convergence of multi-head softmax transformers trained with GD in ICL scenarios. Alternative theoretical perspectives have also been explored: for instance, (Mahankali et al., 2023) demonstrated that a transformer performing a single GD step on a least-squares objective can serve as a global minimizer of the pre-training loss, offering a different interpretation of training objectives in ICL. Similarly, (Ahn et al., 2023) showed that a single-layer model, when trained on random linear regression tasks, implicitly learns to perform a preconditioned GD step at test time, further reinforcing the connection between ICL and optimization-based learning rules. Meanwhile, (Li et al., 2024b; 2025) offered a theoretical interpretation of GLA through the lens of weighted preconditioned GD, although their analysis remains limited to stationary regression settings. Beyond first-order methods, more advanced optimization techniques have also been considered; for example, (Fu et al., 2024) analyzed the convergence behavior of second-order methods in ICL, highlighting their potential for accelerated adaptation relative to first-order approaches.

## 2 IN-CONTEXT LEARNING TIME-VARYING FUNCTIONS

This work builds upon the well-established in-context learning (ICL) framework introduced in (Garg et al., 2022), which aims to train models capable of performing ICL within a specified function class. As discussed in prior work, significant efforts have been devoted to elucidating the mechanisms underlying ICL. In particular, a number of studies (Garg et al., 2022; Akyürek et al., 2022; Mahankali et al.; Ahn et al., 2023; Huang et al., 2024; Zhang et al., 2024; Li et al., 2024b; 2025; Zhang et al., 2025) have investigated the dynamics of ICL in transformer architectures through the lens of linear regression tasks, where the target function is typically assumed to take the form $f(\boldsymbol{x}) = \langle \boldsymbol{w}, \boldsymbol{x} \rangle$. However, these studies commonly rely on the simplifying assumption that the regression weight vector $\boldsymbol{w}$ remains fixed throughout the task. This stationarity assumption creates a theoretical-practical gap, as it does not faithfully reflect real-world scenarios in which data distributions are often non-stationary and the underlying regression weights may vary across different input samples.

**In-context Learning Time-varying Functions** To bridge this gap and advance the theoretical understanding of ICL in non-stationary settings, we introduce a more realistic framework in which the labels in the training prompt are generated by time-varying functions. Formally, let $\mathcal{D}_{\mathcal{X}}$ denote a distribution over inputs and $\mathcal{D}_{\mathcal{F}_i}$ a time-varying distribution over functions in $\mathcal{F}_i$. A prompt $P$ is defined as a sequence $(\boldsymbol{x}_1, f_1(\boldsymbol{x}_1), \ldots, \boldsymbol{x}_n, f_n(\boldsymbol{x}_n), \boldsymbol{x}_{\text{query}})$, where the inputs $\boldsymbol{x}_1, \ldots, \boldsymbol{x}_n \in \mathbb{R}^d$ and query $\boldsymbol{x}_{\text{query}} = \boldsymbol{x}_{n+1} \in \mathbb{R}^d$ are drawn from $\mathcal{D}_{\mathcal{X}}$, and each $f_i$ is drawn from $\mathcal{D}_{\mathcal{F}_i}$. One may consider two canonical types of time-varying functions inspired by the literature:

- *Deterministic time-varying functions*: Here, $f_i = f(\cdot, i/(n+1))$, where $f$ is assumed to vary smoothly over rescaled time. This setting captures gradual and predictable evolution in the underlying mapping, as extensively studied in time-varying nonlinear regression models (Zhang & Wu, 2012; 2015).
- *Stochastic time-varying functions*: In this case, the evolution of $f_i$ is modeled as a stochastic process, allowing for random fluctuations in the function mapping. A representative model is $f_i(x) = \gamma f_{i-1}(x) + e_i(x)$, where $0 < \gamma < 1$ is a forgetting factor modeling gradual drift in task mappings and $e_i(x)$ is a zero-mean stochastic perturbation.

**Definition 1.** *We say that a model $\mathcal{M}$ can in-context learn the time-varying function class $\mathcal{F}_i$ up to accuracy $\epsilon$, with respect to $(\mathcal{F}_i, \mathcal{D}_{\mathcal{X}})$, if it can predict $f_{n+1}(\boldsymbol{x}_{query})$ based on the prompt $P$ with average error*

$$\mathbb{E}_P \left[ \ell(\mathcal{M}(P), f_{n+1}(\boldsymbol{x}_{query})) \right] \leq \epsilon, \tag{1}$$

*where $\ell(\cdot, \cdot)$ denotes an appropriate loss function, such as squared error.*

Within this framework, we then pose the following central question:

      **Question:** Can we train a model to in-context learn a given time-varying function class?

In this work, to facilitate theoretical analysis while preserving non-stationarity, we consider a simple yet expressive instantiation of the function class:

$$y_i = f_i(\boldsymbol{x}_i) = \langle \boldsymbol{w}_i, \boldsymbol{x}_i \rangle \in \mathbb{R}, i \in [n+1], \tag{2}$$

where each weight vector $\boldsymbol{w}_i$ evolves according to a first-order autoregressive process given by

$$\boldsymbol{w}_i = \gamma \boldsymbol{w}_{i-1} + \boldsymbol{e}_i, i \in [n+1]. \tag{3}$$

Here, $0 < \gamma \le 1$ is the autoregressive coefficient that controls the temporal correlation of the weight vectors, the sequence $\boldsymbol{w}_i$ follows a random walk model, which is a widely adopted generative model in signal processing and adaptive filtering literature (Sayed, 2011). To facilitate tractable analysis, we further assume that the initial weight vector is drawn i.i.d. as $\boldsymbol{w}_0 \overset{\text{i.i.d.}}{\sim} \mathcal{N}(\boldsymbol{0}, \sigma_w^2 \mathbf{I})$, the noisy terms are i.i.d. Gaussian with[1] $\boldsymbol{e}_i \overset{\text{i.i.d.}}{\sim} \mathcal{N}(\boldsymbol{0}, \sigma_e^2 \mathbf{I})$, and the input vectors are i.i.d. samples from a zero-mean Gaussian distribution with covariance matrix $\boldsymbol{x}_i \overset{\text{i.i.d.}}{\sim} \mathcal{N}(\boldsymbol{0}, \boldsymbol{\Lambda})$. Moreover, we assume that the random variables $\boldsymbol{w}_{i-1}$, $\boldsymbol{e}_i$, and $\boldsymbol{x}_i$ are mutually independent. Following a long line of theoretical work on in-context learning (Mahankali et al.; Ahn et al., 2023; Zhang et al., 2024; Chen et al., 2024; Yang et al., 2024; Li et al., 2024b; 2025; Zhang et al., 2025), we adopt Gaussian assumptions in our analysis. This modeling choice enables sharp and explicit characterizations of both the training and test errors–rather than only providing loose upper bounds–and is therefore essential for isolating how key quantities such as $\gamma$ govern the behavior of the learned in-context learner.

**Gated Linear Attention**    In the non-stationary regression setting introduced above, where the underlying task weights evolve gradually over time, it is crucial for the model to effectively capture pairwise correlations while adapting to the dynamics of changing tasks. Although standard linear attention mechanisms offer computational efficiency and scalability, they lack the flexibility to modulate the influence of prior context based on its relevance to the current input–an ability that is particularly important in nonstationary environments.

To address this limitation, we employ Gated Linear Attention (GLA) (Yang et al., 2023; Li et al., 2024b; 2025), which enhances linear attention by introducing a gating mechanism that controls the flow of past information. This structure enables the model to selectively integrate relevant historical patterns while suppressing outdated ones, thereby offering a better inductive bias for capturing evolving structures in non-stationary tasks.

Formally, we consider the following implementation of GLA. Let $\boldsymbol{W}_Q \in \mathbb{R}^{(d+1) \times (d+1)}$, $\boldsymbol{W}_K \in \mathbb{R}^{(d+1) \times (d+1)}$, and $\boldsymbol{W}_V \in \mathbb{R}^{(d+1) \times (d+1)}$ denote the query, key, and value weight matrices, respectively. To streamline the subsequent analysis, we follow prior works (Ahn et al., 2023; Huang et al., 2024; Zhang et al., 2024; Li et al., 2024b; 2025; Zhang et al., 2025) and construct the prompt by evaluating each function $f_i$ on the sampled inputs and pairing each input with its corresponding output:

$$\boldsymbol{Z} = \begin{bmatrix} \boldsymbol{z}_1 & \cdots & \boldsymbol{z}_n & \boldsymbol{z}_{n+1} \end{bmatrix} = \begin{bmatrix} \boldsymbol{x}_1 & \cdots & \boldsymbol{x}_n & \boldsymbol{x}_{n+1} \\ y_1 & \cdots & y_n & 0 \end{bmatrix} \in \mathbb{R}^{(d+1) \times (n+1)}. \tag{4}$$

For each input $\boldsymbol{z}_i$, we define the corresponding query, key, and value vectors as $\boldsymbol{q}_i = \boldsymbol{W}_Q \boldsymbol{z}_i$, $\boldsymbol{k}_i = \boldsymbol{W}_K \boldsymbol{z}_i$ and $\boldsymbol{v}_i = \boldsymbol{W}_V \boldsymbol{z}_i$. The output of GLA at position $i$ is given by:

$$\boldsymbol{o}_i = \boldsymbol{S}_i \boldsymbol{q}_i \quad \text{and} \quad \boldsymbol{S}_i = \lambda \boldsymbol{S}_{i-1} + \boldsymbol{v}_i \boldsymbol{k}_i^\top, \tag{5}$$

where $\lambda \in (0, 1]$ is a forgetting factor that determines how quickly the attention mechanism discounts earlier information. For ease of theoretical analysis, we adopt a simplified formulation where a single global forgetting factor $\lambda$ is used, rather than assigning a separate, data-dependent gating coefficient to each token as done in the original GLA model. By unrolling the recursive update in (5), we obtain:

$$\boldsymbol{S}_{n+1} = \lambda \boldsymbol{S}_n + \boldsymbol{v}_{n+1} \boldsymbol{k}_{n+1}^\top = \sum_{i=1}^{n+1} \lambda^{n+1-i} \boldsymbol{v}_i \boldsymbol{k}_i^\top = \boldsymbol{W}_V \left( \sum_{i=1}^{n+1} \lambda^{n+1-i} \boldsymbol{z}_i \boldsymbol{z}_i^\top \right) \boldsymbol{W}_K^\top, \tag{6}$$

which leads to the following expression for the output vector:

$$\boldsymbol{o}_{n+1} = \boldsymbol{S}_{n+1} \boldsymbol{q}_{n+1} = \boldsymbol{W}_V \left( \sum_{i=1}^{n+1} \lambda^{n+1-i} \boldsymbol{z}_i \boldsymbol{z}_i^\top \right) \boldsymbol{W}_K^\top \boldsymbol{W}_Q \boldsymbol{z}_{n+1}. \tag{7}$$

It is worth noting that when $\lambda = 1$, the weighted sum degenerates into an unweighted accumulation, i.e., $\sum_{i=1}^{n+1} \boldsymbol{z}_i \boldsymbol{z}_i^\top = \boldsymbol{Z}\boldsymbol{Z}^\top$, under which the GLA formulation reduces to the standard linear attention

---

[1]One may assume $\sigma_e^2 = (1 - \gamma^2)\sigma_w^2$ to ensure $\mathbb{E}[\|\boldsymbol{w}_i\|^2] = \mathbb{E}[\|\boldsymbol{w}_{i-1}\|^2]$. In this work, we relax this constraint to allow more general settings.

model. This highlights that GLA generalizes linear attention by introducing a learnable memory decay.

Since the final prediction is taken as the last entry of the token vector output by the GLA layer, only a subset of the entries in the weight matrices $\boldsymbol{W}_V$ and $\boldsymbol{W}_Q$, $\boldsymbol{W}_K$ influence the output. To simplify the notation and subsequent analysis, we merge the query and key matrices into a single matrix and define

$$\boldsymbol{W}_V = \begin{bmatrix} \boldsymbol{W}_{11}^V & \boldsymbol{w}_{12}^V \\ \boldsymbol{w}_{21}^{V\top} & w_{-1}^V \end{bmatrix} \in \mathbb{R}^{(d+1)\times(d+1)} \text{ and } \boldsymbol{W}_{KQ} = \begin{bmatrix} \boldsymbol{W}_{11}^{KQ} & \boldsymbol{w}_{12}^{KQ} \\ \boldsymbol{w}_{21}^{KQ\top} & w_{-1}^{KQ} \end{bmatrix} \in \mathbb{R}^{(d+1)\times(d+1)}, \quad (8)$$

where $\boldsymbol{W}_{11}^V, \boldsymbol{W}_{11}^{KQ} \in \mathbb{R}^{d\times d}$, $\boldsymbol{w}_{12}^V, \boldsymbol{w}_{21}^V, \boldsymbol{w}_{12}^{KQ}, \boldsymbol{w}_{21}^{KQ} \in \mathbb{R}^{d\times 1}$ and $w_{-1}^V, w_{-1}^{KQ} \in \mathbb{R}$. Using this decomposition, we express the predicted output as

$$\widehat{y}_{n+1} = \boldsymbol{o}_{n+1}(d+1) = \begin{bmatrix} \boldsymbol{w}_{21}^{V\top} & w_{-1}^V \end{bmatrix} \left( \sum_{i=1}^{n+1} \lambda^{n+1-i} \boldsymbol{z}_i \boldsymbol{z}_i^\top \right) \begin{bmatrix} \boldsymbol{W}_{11}^{KQ} \\ \boldsymbol{w}_{21}^{KQ\top} \end{bmatrix} \boldsymbol{x}_{n+1}. \quad (9)$$

Note that only the last row of $\boldsymbol{W}_V$ and the first $d$ columns of $\boldsymbol{W}_{KQ}$ contribute to the final prediction. Therefore, without loss of generality, we may set the remaining entries in $\boldsymbol{W}_V$ and $\boldsymbol{W}_{KQ}$ to zero in the subsequent analysis.

## 3 THEORETICAL ANALYSIS OF GLA FOR TIME-VARYING REGRESSION

In this work, we investigate the convergence behavior, training error, and testing error of ICL linear predictors based on the GLA model for time-varying functions. Suppose we are given $B$ independent in-context learning training tasks, where each task prompt corresponds to an embedding matrix $\boldsymbol{Z}_\tau$, for $\tau = 1, \ldots, B$, constructed according to the transformation defined in (4):

$$\boldsymbol{Z}_\tau = \begin{bmatrix} \boldsymbol{z}_{\tau,1} & \cdots & \boldsymbol{z}_{\tau,n} & \boldsymbol{z}_{\tau,n+1} \end{bmatrix} = \begin{bmatrix} \boldsymbol{x}_{\tau,1} & \cdots & \boldsymbol{x}_{\tau,n} & \boldsymbol{x}_{\tau,n+1} \\ \langle \boldsymbol{w}_{\tau,1}, \boldsymbol{x}_{\tau,1} \rangle & \cdots & \langle \boldsymbol{w}_{\tau,n}, \boldsymbol{x}_{\tau,n} \rangle & 0 \end{bmatrix}, \quad (10)$$

where the weight vectors $\boldsymbol{w}_{\tau,i}$ evolves according to (3).

We denote the prediction produced by the GLA model on the query input of task $\tau$ as $\widehat{y}_{\tau,n+1}$, whose exact form is given in (9). The empirical risk over $B$ independent task prompts is then defined as:

$$l(\boldsymbol{\theta}) = \frac{1}{2B} \sum_{\tau=1}^B \left( \widehat{y}_{\tau,n+1} - \langle \boldsymbol{w}_{\tau,n+1}, \boldsymbol{x}_{\tau,n+1} \rangle \right)^2, \quad (11)$$

where the model parameters are denoted by $\boldsymbol{\theta} = \{\boldsymbol{W}_{KQ}, \boldsymbol{W}_V\}$. To analyze the learning dynamics, we consider the population risk induced in the limit as the number of training prompts tends to infinity, i.e., $B \to \infty$:

$$L(\boldsymbol{\theta}) = \lim_{B \to \infty} l(\boldsymbol{\theta}) = \frac{1}{2} \mathbb{E}_{\boldsymbol{w}_{n+1}, \boldsymbol{x}_{n+1}}[(\widehat{y}_{n+1} - \langle \boldsymbol{w}_{n+1}, \boldsymbol{x}_{n+1} \rangle)^2], \quad (12)$$

where we omit the task index $\tau$ for notational simplicity.

We study the evolution of the model parameters under gradient flow, which characterizes the continuous-time limit of gradient descent with infinitesimal step sizes. The parameter dynamics are governed by the ordinary differential equation $\frac{d\boldsymbol{\theta}}{dt} = -\nabla L(\boldsymbol{\theta})$. To facilitate the analysis, following the approach of (Zhang et al., 2024), we introduce an initialization scheme that satisfies the following assumption.

**Assumption 1.** *(Initialization) Let $\sigma > 0$ be a parameter and $\boldsymbol{\Theta} \in \mathbb{R}^{d\times d}$ be any matrix satisfying $\|\boldsymbol{\Theta}\boldsymbol{\Theta}^\top\|_F = 1$ and $\boldsymbol{\Lambda}\boldsymbol{\Theta} \neq \boldsymbol{0}_{d\times d} \in \mathbb{R}^{d\times d}$. We assume*

$$\boldsymbol{W}_V(0) = \sigma \begin{bmatrix} \boldsymbol{0}_{d\times d} & \boldsymbol{0}_d \\ \boldsymbol{0}_d^\top & 1 \end{bmatrix} \in \mathbb{R}^{(d+1)\times(d+1)} \text{ and } \boldsymbol{W}_{KQ}(0) = \sigma \begin{bmatrix} \boldsymbol{\Theta}\boldsymbol{\Theta}^\top & \boldsymbol{0}_d \\ \boldsymbol{0}_d^\top & 0 \end{bmatrix} \in \mathbb{R}^{(d+1)\times(d+1)}. \ (13)$$

Under this setup, the following result establishes that the gradient flow dynamics with respect to the population loss converge to a specific global optimum.

**Theorem 1.** *(Convergence of gradient flow) Consider gradient flow over the population loss in* (12). *Assume that* $\gamma < 1$, *the initial task weight* $\boldsymbol{w}_0 \overset{i.i.d.}{\sim} \mathcal{N}(\mathbf{0}, \sigma_w^2 \mathbf{I})$, *noises* $\boldsymbol{e}_i \overset{i.i.d.}{\sim} \mathcal{N}(\mathbf{0}, \sigma_e^2 \mathbf{I})$ *and inputs* $\boldsymbol{x}_i \overset{i.i.d.}{\sim} \mathcal{N}(\mathbf{0}, \boldsymbol{\Lambda})$. *Suppose the initialization satisfies Assumption 1 with initialization scale* $\sigma > 0$ *satisfying* $\sigma < \sqrt{\frac{2D_1}{\sqrt{d}\|\widetilde{\boldsymbol{\Lambda}}\|}}$ *where*

$$
D_1 = \begin{cases} \lambda^{2n+2} n \sigma_w^2 + \left( \frac{\lambda^2(1-\lambda^{2n})}{(1-\lambda^2)^2} - \frac{\lambda^{2n+2}}{1-\lambda^2} n \right) \sigma_e^2, & \lambda = \gamma, \\ \frac{\lambda^{n+1}\gamma^{n+2} - \lambda\gamma^{2n+2}}{\lambda - \gamma} \sigma_w^2 + \left( \frac{\lambda\gamma(1-\lambda^n\gamma^n)}{(1-\gamma^2)(1-\lambda\gamma)} - \frac{\lambda^{n+1}\gamma^{n+2} - \lambda\gamma^{2n+2}}{(\lambda-\gamma)(1-\gamma^2)} \right) \sigma_e^2, & \lambda \neq \gamma, \end{cases}
$$

*and* $\widetilde{\boldsymbol{\Lambda}} = D_2(2\boldsymbol{\Lambda} + \mathrm{trace}(\boldsymbol{\Lambda})\mathbf{I}) + D_3\boldsymbol{\Lambda}$ *with*

$$
D_2 = \begin{cases} \lambda^{2n+2} n \sigma_w^2 - \left( \frac{n\lambda^{2n+2}}{1-\lambda^2} - \frac{\lambda^4 - \lambda^{2n+2}}{(1-\lambda^2)^2} \right) \sigma_e^2, & \lambda = \gamma, \\ \frac{\gamma^2\lambda^{2n+2} - \lambda^2\gamma^{2n+2}}{\lambda^2 - \gamma^2} \sigma_w^2 - \left( \frac{\gamma^2\lambda^{2n+2} - \lambda^2\gamma^{2n+2}}{(\lambda^2-\gamma^2)(1-\gamma^2)} - \frac{\lambda^2 - \lambda^{2n+2}}{(1-\gamma^2)(1-\lambda^2)} \right) \sigma_e^2, & \lambda \neq \gamma, \end{cases}
$$

*and*

$$
D_3 = \begin{cases} \lambda^{2n+2} n(n-1)\sigma_w^2 + \left( \frac{2n(\lambda^4 - \lambda^{2n+2})}{(1-\lambda^2)^2} - \frac{2(\lambda^{2n+4} - n\lambda^6 + (n-1)\lambda^4)}{(1-\lambda^2)^3} - \frac{\lambda^{2n+2}n(n-1)}{1-\lambda^2} \right)\sigma_e^2, & \lambda = \gamma, \\ \left( \frac{2\gamma^3\lambda^{2n+3} - 2\lambda^5\gamma^{2n+1}}{\lambda(\lambda-\gamma)^2(\lambda+\gamma)} - \frac{2\gamma^{n+2}\lambda^{n+2} - 2\gamma^{2n+1}\lambda^3}{\lambda-\gamma} \right)\sigma_w^2 + \left( \frac{2\gamma^{-1}(\lambda^4 - \lambda^{2n+2})}{(1-\gamma^2)(\lambda-\gamma)(1-\lambda^2)} \right. \\ \quad \left. - \frac{2(\lambda^3 - \lambda^{n+2}\gamma^{n-1})}{(1-\gamma^2)(\lambda-\gamma)(1-\lambda\gamma)} - \frac{2\gamma^3\lambda^{2n+3} - 2\lambda^5\gamma^{2n+1}}{\lambda(\lambda-\gamma)^2(\lambda+\gamma)(1-\gamma^2)} + \frac{2\gamma^{n+2}\lambda^{n+2} - 2\gamma^{2n+1}\lambda^3}{(\lambda-\gamma)(1-\gamma^2)} \right)\sigma_e^2, & \lambda \neq \gamma. \end{cases}
$$

*Then gradient flow converges to a global minimum of the population loss* (12). *Moreover,* $\boldsymbol{W}_{KQ}(0)$ *and* $\boldsymbol{W}_V(0)$ *respectively converge to*

$$
\lim_{t\to\infty} \boldsymbol{W}_V(t) = \sqrt{D_1 \|\widetilde{\boldsymbol{\Lambda}}^{-1}\|_F} \begin{bmatrix} \mathbf{0}_{d\times d} & \mathbf{0}_d \\ \mathbf{0}_d^\top & 1 \end{bmatrix} \quad and \quad \lim_{t\to\infty} \boldsymbol{W}_{KQ}(t) = \sqrt{D_1 \|\widetilde{\boldsymbol{\Lambda}}^{-1}\|_F^{-1}} \begin{bmatrix} \widetilde{\boldsymbol{\Lambda}}^{-1} & \mathbf{0}_d \\ \mathbf{0}_d^\top & 0 \end{bmatrix}. \quad (14)
$$

The proof is deferred to Appendix B. Despite the non-stationary nature of the regression model considered in this work, we establish that gradient flow converges to a global minimum even under random initialization. The closed-form solution in (14) reveals that the location of the global optimum is explicitly determined by $\lambda$ and $\gamma$, highlighting their structural influence on the solution. While the main theorem focuses on the regime $0 < \lambda \leq 1$ and $0 < \gamma < 1$, a more general result accommodating arbitrary $\lambda > 0$ and $\gamma > 0$ is established in Theorem 4 of Appendix B. Moreover, in the limiting case where $\lambda = \gamma = 1$ and $\sigma_e^2 = 0$, the result reduces precisely to that in (Zhang et al., 2024, Theorem 4), thereby recovering the stationary setting as a special case of our more general formulation.

**Training error** We now analyze the training error of the learned network. At the global optimum–i.e., when the parameters converge to $\lim_{t\to\infty} \boldsymbol{W}_V(t)$ and $\lim_{t\to\infty} \boldsymbol{W}_{KQ}(t)$ in (14), a straightforward calculation yields the prediction $\widehat{y}_{n+1}$ as follows:

$$
\widehat{y}_{n+1} = D_1 \begin{bmatrix} \mathbf{0}_d^\top & 1 \end{bmatrix} \left( \sum_{i=1}^{n+1} \lambda^{n+1-i} \boldsymbol{z}_i \boldsymbol{z}_i^\top \right) \begin{bmatrix} \widetilde{\boldsymbol{\Lambda}}^{-1} \\ \mathbf{0}_d^\top \end{bmatrix} \boldsymbol{x}_{n+1} = D_1 \left( \sum_{i=1}^n \lambda^{n+1-i} \boldsymbol{w}_i^\top \boldsymbol{x}_i \boldsymbol{x}_i^\top \right) \widetilde{\boldsymbol{\Lambda}}^{-1} \boldsymbol{x}_{n+1}. \quad (15)
$$

This expression confirms that, for sufficiently long prompts, the trained model successfully in-context learns the family of linear predictors. We emphasize that both $\lambda$ and $\gamma$ jointly influence the degree of time variation in the underlying model. We next quantify the training error at the global optimum.

**Theorem 2.** *(Training error) Assuming the conditions in Theorem 1 hold, the recovery error between* (15) *and* (2) *is*

$$
\mathbb{E}[(\widehat{y}_{n+1} - y_{n+1})^2] = D_1^2 \, \mathrm{trace}\left( D_2(\boldsymbol{\Lambda}\,\mathrm{trace}(\widetilde{\boldsymbol{\Lambda}}^{-1}\boldsymbol{\Lambda}\widetilde{\boldsymbol{\Lambda}}^{-1}\boldsymbol{\Lambda}) + 2\boldsymbol{\Lambda}\widetilde{\boldsymbol{\Lambda}}^{-1}\boldsymbol{\Lambda}\widetilde{\boldsymbol{\Lambda}}^{-1}\boldsymbol{\Lambda} \right)
$$
$$
+ D_3\boldsymbol{\Lambda}\widetilde{\boldsymbol{\Lambda}}^{-1}\boldsymbol{\Lambda}\widetilde{\boldsymbol{\Lambda}}^{-1}\boldsymbol{\Lambda}) + D_4\,\mathrm{trace}(\boldsymbol{\Lambda}) - 2D_1^2\,\mathrm{trace}(\boldsymbol{\Lambda}\widetilde{\boldsymbol{\Lambda}}^{-1}\boldsymbol{\Lambda}), \quad (16)
$$

*where* $D_4 = \gamma^{2n+2}\sigma_w^2 + \frac{1-\gamma^{2n+2}}{1-\gamma^2}\sigma_e^2$.

The proof is provided in Appendix C. Equation (16) illustrates that the training error depends jointly on the parameters $\lambda$ and $\gamma$. Consequently, for fixed $\gamma$, there exists an optimal value of $\lambda$

that minimizes the error. Although the expressions of $D_i$ suggest a symmetric structure in $\lambda$ and $\gamma$, it does not necessarily imply that choosing $\lambda = \gamma$ minimizes the recovery error. In fact, the error involves a subtle balance between the $\sigma_w^2$- and $\sigma_e^2$-dependent terms as well as the trace terms with $\widetilde{\Lambda}^{-1}$. When $\lambda = \gamma$, the simplification of $D_i$ may amplify certain noise-dependent factors and deteriorate the overall error. This observation highlights that the optimal choice of $\lambda$ depends not only on the apparent algebraic symmetry but also on the interplay between noise statistics, system dimension, and the spectral structure of $\Lambda$.

We now consider a special case with $\Lambda = I$, in which the training error in (16) reduces to $\mathbb{E}[(\widehat{y}_{n+1} - y_{n+1})^2] = dD_4 - \frac{dD_1^2}{(2+d)D_2+D_3}$. The first term $dD_4$ only depends on the autoregressive process and is independent to the choice of $\lambda$. To study how the second term varies according to $\lambda$, assume $\gamma < 1$ is fixed and $d$ is sufficiently large, such that $\frac{dD_1^2}{(2+d)D_2+D_3} \approx D_1^2/D_2 =: \xi(\lambda)$. To further illustrate how $\xi(\lambda)$ might vary with $\lambda$, we consider two particular cases. Case I: $\sigma_e \ll \sigma_w$. In this regime, $D_1$ and $D_2$ are respectively dominated by $\frac{\lambda^{n+1}\gamma^{n+2}-\lambda\gamma^{2n+2}}{\lambda-\gamma}\sigma_w^2$ and $\frac{\gamma^2\lambda^{2n+2}-\lambda^2\gamma^{2n+2}}{\lambda^2-\gamma^2}\sigma_w^2$. Consequently, $\xi(\lambda)$ can be well approximated by $\frac{\gamma^{2n+4}(\lambda^n-\gamma^n)(\lambda+\gamma)}{(\lambda^n+\gamma^n)(\lambda-\gamma)}\sigma_w^2$, which increases monotonically with $\lambda$ on $(0, \gamma)$ and then decreases monotonically on $(\gamma, 1)$. Case II: $n$ is sufficiently large such that $D_1$ and $D_2$ are dominated by $\frac{\lambda\gamma}{(1-\gamma^2)(1-\lambda\gamma)}\sigma_e^2$ and $\frac{\lambda^2-\lambda^{2n+2}}{(1-\gamma^2)(1-\lambda^2)}\sigma_e^2$, respectively. In this case, $\xi(\lambda) \approx \frac{\gamma^2(1-\lambda^2)}{(1-\gamma^2)(1-\lambda\gamma)^2(1-\lambda^{2n})}\sigma_e^2$, which likewise increases monotonically with $\lambda$ on $(0, \gamma)$ and then decreases monotonically on $(\gamma, 1)$. Although these results are derived under simplifying approximations, they suggest that the training loss exhibits an inverse U-shaped dependence on $\lambda$, attaining its minimum at some $\lambda < 1$ rather than at $\lambda = 1$. The subsequent experiments provide direct validation of these theoretical predictions.

**Testing error**   In this part, we characterize the prediction performance of the trained transformer when evaluated on a test prompt drawn from a potentially different task distribution. Notably, the model parameters are fixed at their global optimum obtained from training, and the test prompt may differ in its length, data distribution, and underlying dynamics. We consider test prompts of the form

$$
\overline{Z} = \begin{bmatrix} \overline{z}_1 & \cdots & \overline{z}_m & \overline{z}_{m+1} \end{bmatrix} = \begin{bmatrix} \overline{x}_1 & \cdots & \overline{x}_m & \overline{x}_{m+1} \\ \overline{y}_1 & \cdots & \overline{y}_m & 0 \end{bmatrix}
$$

$$
= \begin{bmatrix} \overline{x}_1 & \cdots & \overline{x}_m & \overline{x}_{m+1} \\ \langle \overline{w}_1, \overline{x}_1 \rangle & \cdots & \langle \overline{w}_m, \overline{x}_m \rangle & 0 \end{bmatrix}, \tag{17}
$$

where the latent task weights $\{\overline{w}_i\}_{i=1}^{m+1}$ evolve according to the first-order autoregressive model $\overline{w}_i = \overline{\gamma} \cdot \overline{w}_{i-1} + \overline{e}_i, i = 1, \ldots, m+1$. To distinguish between training and testing distributions, we assume that the initial weight vector satisfies $\overline{w}_0 \overset{\text{i.i.d.}}{\sim} \mathcal{N}(0, \overline{\sigma}_w^2 I)$, and the driving noise $\overline{e}_i \overset{\text{i.i.d.}}{\sim} \mathcal{N}(0, \overline{\sigma}_e^2 I)$. The inputs are drawn independently as $\overline{x}_i \overset{\text{i.i.d.}}{\sim} \mathcal{N}(0, \overline{\Lambda})$, and we assume mutual independence among random variables $\overline{w}_{i-1}, \overline{e}_i$, and $\overline{x}_i$.

Given a forgetting factor $\overline{\lambda}$, the prediction $\widetilde{y}_{m+1}$ produced by the model at test time (evaluated at the training global optimum) is

$$
\widetilde{y}_{m+1} = D_1 \Big( \sum_{i=1}^m \overline{\lambda}^{m+1-i} \overline{w}_i^\top \overline{x}_i \overline{x}_i^\top \Big) \widetilde{\Lambda}^{-1} \overline{x}_{m+1}. \tag{18}
$$

We now characterize the mean squared prediction error on the test prompt:

**Theorem 3.** *(Testing error) Under the assumptions in Theorem 4, the expected prediction error of the model on the test prompt is given by*

$$
\mathbb{E}[(\widetilde{y}_{m+1} - \overline{y}_{m+1})^2] = D_1^2 \operatorname{trace}\big(\overline{D}_2(\overline{\Lambda}\operatorname{trace}(\widetilde{\Lambda}^{-1}\overline{\Lambda}\widetilde{\Lambda}^{-1}\overline{\Lambda}) + 2\overline{\Lambda}\widetilde{\Lambda}^{-1}\overline{\Lambda}\widetilde{\Lambda}^{-1}\overline{\Lambda})
$$
$$
+ \overline{D}_3\overline{\Lambda}\widetilde{\Lambda}^{-1}\overline{\Lambda}\widetilde{\Lambda}^{-1}\overline{\Lambda}\big) + \overline{D}_4 \operatorname{trace}(\overline{\Lambda}) - 2D_1 \cdot \overline{D}_1 \operatorname{trace}(\overline{\Lambda}\widetilde{\Lambda}^{-1}\overline{\Lambda}), \tag{19}
$$

*where $\overline{D}_i$ for $i = 1, \ldots, 4$ are defined analogously to the $D_i$ constants from training, with the substitution $\lambda \to \overline{\lambda}$, $\gamma \to \overline{\gamma}$, $\sigma_w^2 \to \overline{\sigma}_w^2$, $\sigma_e^2 \to \overline{\sigma}_e^2$, and $n \to m$.*

The proof has been provided in Appendix D. This result quantifies the generalization behavior of the trained model when applied to unseen prompts sampled from a potentially different distribution.

Notably, the prediction error depends jointly on the training and testing task statistics through the interaction between $\widetilde{\Lambda}$ and $\overline{\Lambda}$. Moreover, the expected error $\mathbb{E}[(\widetilde{y}_{m+1} - \overline{y}_{m+1})^2]$ is inherently nonzero due to the stochastic nature of the task evolution–specifically, the noise in the dynamics of $\overline{w}_i$ introduces irreducible uncertainty in the test labels $\overline{y}_i$. This highlights the importance of employing GLA, which adaptively modulates the influence of past observations and better accommodates temporal variations in the underlying regression weights. In the subsequent experimental section, we empirically demonstrate the effectiveness of the GLA mechanism in handling non-stationary tasks.

**Comparison with Adaptive Signal Processing**    The non-stationary regression setting considered in this paper is closely related to classical problems in adaptive signal processing, where the underlying model parameters evolve gradually over time (Sayed, 2011; Das et al., 2015; Abdolee et al., 2016; Qin et al., 2020; Claser & Nascimento, 2021; Yu et al., 2021; Wang et al., 2022). To track such non-stationary dynamics, a wide range of online algorithms have been developed, including the least mean squares (LMS) algorithm, the affine projection algorithm (APA), and the recursive least squares (RLS) algorithm. These methods are designed to update model parameters iteratively in response to streaming data, with the goal of minimizing instantaneous or long-term prediction error. Under non-stationary models such as the first-order autoregressive process described in (3), the corresponding theoretical error analyses for these methods also indicate that, for a fixed $\gamma$, there exists an optimal choice of step size (in LMS/APA) or forgetting factor (in RLS) that minimizes the tracking error.

While classical adaptive signal processing methods explicitly update model parameters over time based on streaming observations, the paradigm studied in this paper–in-context learning with the GLA model–adopts a fundamentally different approach. Instead of relying on explicit parameter updates, as in LMS, APA, or RLS, the GLA implicitly adapts to task dynamics via internal representations conditioned on the prompt. In particular, the gating mechanism in GLA enables the model to selectively integrate past information in a soft and differentiable manner, thereby tracking non-stationary structures without modifying its parameters. This architectural distinction offers a new perspective on learning in non-stationary environments, where adaptation arises not from external optimization procedures, but from the model's forward computation itself.

## 4    EXPERIMENTAL RESULTS

In this section, we present experiments to validate the theoretical analysis and demonstrate the advantages of GLA in non-stationary models. The experiments are conducted under the following settings. The training and testing losses are defined as $\frac{1}{B}\sum_{\tau=1}^{B}(\widehat{y}_{\tau,n+1} - y_{\tau,n+1})^2$ and $(\widetilde{y}_{m+1} - \overline{y}_{m+1})^2$, respectively. Unless otherwise specified, we set $d = 10$, $n = 100$, $\sigma_w^2 = 1$, $\sigma_e^2 = 0.01$, and $B = 10^7$. The AdamW optimizer is adopted with learning rate $10^{-2}$, weight decay $0.05$, and momentum parameter $0.9$. Each model is trained for 2000 epochs with a batch size of 5000 samples. The loss associated with the optimal $\lambda$ is highlighted by a star. Although the theoretical analysis imposes a constraint on the initialization matrix, our experiments use a random Gaussian initialization and still observe the predicted behavior, indicating that the constraint is not necessary in practice.

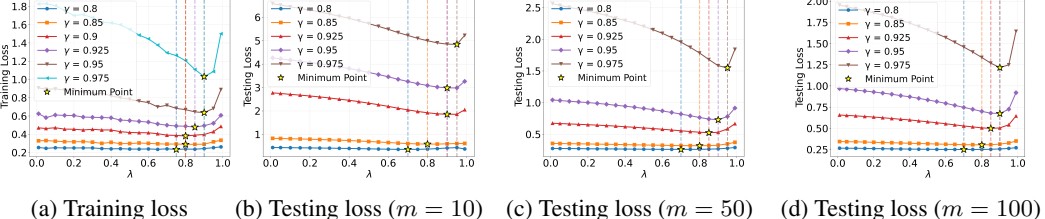

(a) Training loss        (b) Testing loss ($m = 10$)    (c) Testing loss ($m = 50$)    (d) Testing loss ($m = 100$)

Figure 1: Training and testing performance of the one-layer GLA model with different $\lambda$ and $\gamma$.

The first experiment compares the training and testing performance of the one-layer GLA model under varying choices of $\gamma$ and $\lambda$. As shown in Figure 1a, when the autoregressive coefficient $\gamma$ decreases and the impact of noise becomes more pronounced, an appropriate choice of $\lambda$ is required to attain the lowest training loss. During testing, we evaluate the GLA model trained with $\lambda = 0.9$ under different sequence lengths $m \in \{10, 50, 100\}$. The results in Figures 1b to 1d show that, across different values of $\gamma$, selecting an appropriate $\lambda$ remains crucial for minimizing the test loss.

These results highlight the role of GLA in stabilizing learning under non-stationary conditions. By introducing a gating mechanism into linear attention, GLA effectively regulates the influence of past inputs, thereby mitigating error accumulation and enhancing the model's adaptability to distributional shifts. Consequently, GLA achieves longer effective memory and improved generalization, underscoring its advantage in handling time-varying data. As mentioned previously, a one-layer GLA model applied to a first-order autoregressive process functions analogously to an adaptive filter. To illustrate this, we compare its performance with LMS and RLS algorithms. We set the LMS step size to 0.01 and the RLS forgetting factor to 0.98, train on sequences of length 1000, and perform 10,000 Monte Carlo trials, averaging the results. The training errors for LMS and RLS are respectively [0.2639 0.3168 0.6058 1.0072 1.4758] and [0.2555 0.3746 0.6658 0.8881 1.2916] for $\gamma = [0.8\ 0.85\ 0.925\ 0.95\ 0.975]$. Compared to LMS and RLS, which require fixed or slowly adapting parameters, a one-layer GLA model achieves lower training errors (see Figure 1a) because it possesses higher representational flexibility. Furthermore, LMS and RLS adapt only to a single sequence at a time, requiring retraining for each new input, and therefore cannot leverage cross-sequence information. In contrast, GLA's learnable weights are shared across sequences, allowing the model to generalize and adapt efficiently to new inputs without retraining.

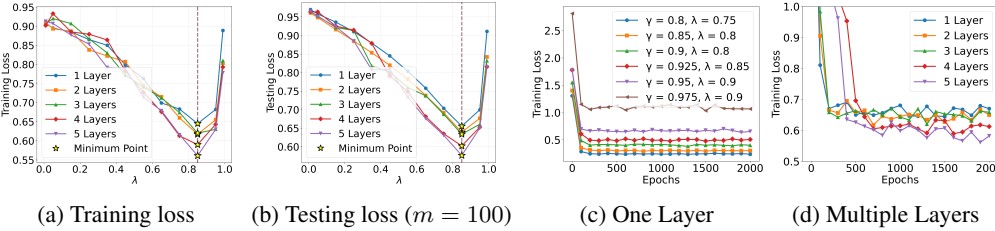

(a) Training loss     (b) Testing loss ($m = 100$)     (c) One Layer     (d) Multiple Layers

Figure 2: (a-b) Training and testing performance of the multi-layer GLA model with $\gamma = 0.95$ and different $\lambda$; (c) convergence performance of the one-layer GLA model; (d) convergence performance for GLA models with different layers.

In the second experiment, we investigate the impact of network depth on the performance of the GLA model. As illustrated in Figures 2a and 2b, increasing the number of layers consistently enhances both training and testing performance, suggesting that deeper architectures can more effectively capture long-range dependencies in non-stationary sequences. Conceptually, each GLA layer implements a linear adaptive filter whose effective behavior is determined by its gating weights. When multiple layers are stacked, these adaptive filters operate at different timescales, enabling the network to simultaneously capture short-term fluctuations and longer-term trends in the evolving regression weights. This multi-timescale structure explains why deeper GLA models achieve better performance under non-stationary regression: a single layer can track only one effective timescale of drift, while multiple layers collectively approximate a richer family of dynamic predictors. While formal theoretical analysis for multi-layer GLA models is not yet established, the empirical results underscore the critical role of the adaptive gating mechanism in regulating information flow across layers, thereby mitigating error accumulation and improving generalization.

Under the same experimental settings as the first two experiments, we examine the training convergence of the one-layer GLA with the optimal $\lambda$ corresponding to the minimum loss, and of the multi-layer GLA with $\lambda = 0.85$. With random Gaussian initialization and a sufficiently large number of training samples, Figure 2c shows that the one-layer GLA achieves linear convergence, in agreement with our previous analysis. Figure 2d further demonstrates that the multi-layer GLA maintains linear convergence, indicating that the adaptive gating mechanism effectively stabilizes gradient propagation across layers. A rigorous theoretical characterization of convergence for multi-layer GLA is left for future work.

In the third experiment, we assess the ICL capability of GLA and Linear Attention (LA) models on a real-world language task. We focus on sentiment classification using the SST-2 dataset (Socher et al., 2013), which contains 67,349 training samples and 872 validation samples with binary labels (positive/negative). To initialize the models, we employ GPT-2 (small) (Radford et al., 2019), which consists of 12 layers, a hidden size of 768, 12 attention heads, and approximately 117M parameters. We then replace the original softmax attention with (i) linear attention, resulting in LinearGPT2, and (ii) gated linear attention, resulting in GatedLinearGPT2. Both models are optimized using AdamW

with a learning rate of $5 \times 10^{-5}$, weight decay of $0.05$, and momentum parameter of $0.9$ for $1,000$ iterations. For ICL fine-tuning, we provide 20 in-context demonstrations per instance, computing the loss only on label tokens. During evaluation on the SST-2 validation set, we vary the number of demonstrations $K \in \{1, 5, 10, 15, 20\}$. Performance is assessed using two metrics: (1) Accuracy, defined as the standard prediction accuracy; and (2) Confidence, calculated for each correctly classified example by converting the model's logits over positive, negative to probabilities $(p_{\text{pos}}, p_{\text{neg}})$ and taking $\max(p_{\text{pos}}, p_{\text{neg}})$, with the reported value being the average over all correctly classified examples. As shown in Figures 3a and 3b, when $\lambda = 0.9$, GLA achieves the highest accuracy and confidence, outperforming LA by a clear margin. This empirical advantage can be attributed to its gating mechanism: unlike LA, which implicitly assumes a stationary linear regression structure, GLA is able to adapt to the non-stationarity of real-world data by selectively integrating or discarding historical information–an ability that proves critical for reliable prediction.

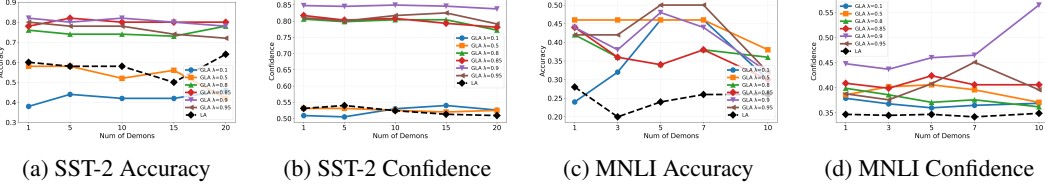

| (a) SST-2 Accuracy | (b) SST-2 Confidence | (c) MNLI Accuracy | (d) MNLI Confidence |

Figure 3: Accuracy and confidence of GatedLinearGPT2 vs. LinearGPT2 on SST-2 sentiment classification (left two) and MNLI natural language inference (right two) across different numbers of demonstrations.

In the final experiment, we evaluate the ICL capabilities of GLA and Linear Attention (LA) models on a more challenging natural language inference task, which requires determining the logical relationship between a premise–hypothesis pair (entailment, contradiction, or neutral) across a broad range of text genres. We use the Multi-Genre Natural Language Inference (MNLI) dataset (Williams et al., 2018), which spans multiple genres and contains approximately 393k training examples with three class labels. Following the same setup as in the third experiment, we provide 10 in-context demonstrations per instance for ICL fine-tuning—constrained by context length—and compute the loss only on the label tokens. For evaluation on the MNLI validation set, we vary the number of demonstrations $K \in \{1, 3, 5, 7, 10\}$. As shown in Figures 3c and 3d, GLA consistently achieves higher accuracy and confidence than LA, highlighting the benefit of the gating mechanism.

## 5 CONCLUSION

This work presents a theoretical investigation of in-context learning in non-stationary regression problems, addressing an important gap in the current understanding of transformer models. Under a first-order autoregressive model of non-stationarity, we show that GLA outperforms standard linear attention by dynamically reweighting past inputs, enabling more accurate prediction in time-varying settings. Our analysis provides rigorous justification for the advantage of gating in capturing distributional shifts and highlights its role as an architectural inductive bias in adaptive learning. These findings not only deepen the theoretical foundations of ICL in dynamic environments but also suggest broader implications for the design of transformer variants in real-world applications characterized by non-stationarity.

A natural direction for future work is to generalize the first-order autoregressive assumption to a broader class of dynamic-weight models. In particular, allowing more flexible temporal evolutions–such as higher-order dynamics, stochastic drift, or slowly varying adversarial changes–would further illuminate how in-context learning behaves in general non-stationary settings. A second direction for future work is to develop a rigorous theoretical characterization of how gating mechanisms interact across multiple GLA layers. While our experiments show that stacking layers consistently improves performance, a principled analysis of how multi-layer structures capture multiple timescales of drift remains an important open problem. The third future direction is to analyze the global optimization landscape of the GLA model studied in this paper. Our numerical experiments suggest that random Gaussian initialization consistently converges to a global minimum under gradient flow, even when the theoretical initialization conditions are violated. This indicates the existence of a benign global optimum and motivates a deeper theoretical study of the model's optimization landscape.

ACKNOWLEDGMENTS

The work has been supported in part by NSF grants IIS-2312840 and IIS-2402952. ZQ gratefully acknowledges support from the MICDE Research Scholars Program at the University of Michigan.

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

# Appendices

## A  TECHNICAL TOOLS USED IN THE PROOFS

**Lemma 1.** *Suppose $\boldsymbol{w}_i = \gamma \boldsymbol{w}_{i-1} + \boldsymbol{e}_i$ where $\boldsymbol{w}_0 \stackrel{i.i.d.}{\sim} \mathcal{N}(\boldsymbol{0}, \sigma_w^2 \mathbf{I})$, $\boldsymbol{e}_i \stackrel{i.i.d.}{\sim} \mathcal{N}(\boldsymbol{0}, \sigma_e^2 \mathbf{I})$ and they are mutually independent. We have*

$$\mathbb{E}[\boldsymbol{w}_a \boldsymbol{w}_b^\top] = \begin{cases} (\sigma_w^2 + \min\{a,b\}\sigma_e^2)\mathbf{I}, & \gamma = 1, \\ (\gamma^{a+b}\sigma_w^2 + \frac{\gamma^{a+b} - \gamma^{|a-b|}}{\gamma^2 - 1}\sigma_e^2)\mathbf{I}, & \gamma \neq 1. \end{cases} \tag{20}$$

*Proof.* For any $a$ and $b$, we have

$$\boldsymbol{w}_a = \gamma^a \boldsymbol{w}_0 + \sum_{i=1}^{a} \gamma^{a-i} \boldsymbol{e}_i, \tag{21}$$

$$\boldsymbol{w}_b = \gamma^b \boldsymbol{w}_0 + \sum_{i=1}^{b} \gamma^{b-i} \boldsymbol{e}_i. \tag{22}$$

Then we have

$$\mathbb{E}[\boldsymbol{w}_a \boldsymbol{w}_b^\top] = \mathbb{E}[\gamma^{a+b} \boldsymbol{w}_0 \boldsymbol{w}_0^\top] + \sum_{j=1}^{\min\{a,b\}} \gamma^{a+b-2j} \, \mathbb{E}[\boldsymbol{e}_j \boldsymbol{e}_j^\top]$$

$$= \gamma^{a+b}\sigma_w^2 \mathbf{I} + \sum_{j=1}^{\min\{a,b\}} \gamma^{a+b-2j} \sigma_e^2 \mathbf{I}$$

$$= \begin{cases} (\sigma_w^2 + \min\{a,b\}\sigma_e^2)\mathbf{I}, & \gamma = 1, \\ (\gamma^{a+b}\sigma_w^2 + \frac{\gamma^{a+b} - \gamma^{|a-b|}}{\gamma^2 - 1}\sigma_e^2)\mathbf{I}, & \gamma \neq 1. \end{cases} \tag{23}$$

$\square$

According to (Zhang et al., 2024, Lemma 5.1), we have

**Lemma 2.** *Let $\boldsymbol{z}_i, i = 1, \ldots, n+1$ be prompts defined in (4). Then the prediction $\widehat{y}_{n+1}$ can be written as a quadratic function as follows:*

$$\widehat{y}_{n+1} = \boldsymbol{u}^\top \boldsymbol{H} \boldsymbol{u}, \tag{24}$$

*where $\boldsymbol{H} = \frac{1}{2}\boldsymbol{X} \otimes \left( \sum_{i=1}^{n+1} \lambda^{n+1-i} \boldsymbol{z}_i \boldsymbol{z}_i^\top \right)$ with $\boldsymbol{X} = \begin{bmatrix} \boldsymbol{0}_{d\times d} & \boldsymbol{x}_{n+1} \\ \boldsymbol{x}_{n+1}^\top & 0 \end{bmatrix} \in \mathbb{R}^{(d+1)\times(d+1)}$ and $\boldsymbol{u} =$ $\mathrm{vec}(\boldsymbol{U}) \in \mathbb{R}^{(d+1)^2 \times 1}$, $\boldsymbol{U} = \begin{bmatrix} \boldsymbol{U}_{11} & \boldsymbol{u}_{12} \\ \boldsymbol{u}_{21}^\top & u_{-1} \end{bmatrix} \in \mathbb{R}^{(d+1)\times(d+1)}$ with $\boldsymbol{U}_{11} = \boldsymbol{W}_{11}^{KQ}$, $\boldsymbol{u}_{21} = \boldsymbol{w}_{21}^{KQ}$, $\boldsymbol{u}_{12} = \boldsymbol{w}_{21}^V$ and $u_{-1} = w_{-1}^V$ defined in (8).*

**Lemma 3.** *Let $\boldsymbol{u} = \mathrm{vec}(\boldsymbol{U}) = \mathrm{vec}\left( \begin{bmatrix} \boldsymbol{U}_{11} & \boldsymbol{u}_{12} \\ \boldsymbol{u}_{21}^\top & u_{-1} \end{bmatrix} \right)$ as Lemma 2. Consider gradient flow over*

$$L(\boldsymbol{u}) = \frac{1}{2} \mathbb{E}[(\boldsymbol{u}^\top \boldsymbol{H} \boldsymbol{u} - \boldsymbol{w}_{n+1}^\top \boldsymbol{x}_{n+1})^2] \tag{25}$$

*with respect to $\boldsymbol{u}$ starting from an initial value satisfying Assumption 1. Then the dynamics of $\boldsymbol{U}$ satisfies*

$$\frac{d\boldsymbol{U}_{11}(t)}{dt} = -u_{-1}^2 \widetilde{\boldsymbol{\Lambda}} \boldsymbol{\Lambda} \boldsymbol{U}_{11} \boldsymbol{\Lambda} + D_1 u_{-1} \boldsymbol{\Lambda}^2, \tag{26}$$

$$\frac{du_{-1}(t)}{dt} = -\mathrm{trace}(u_{-1} \widetilde{\boldsymbol{\Lambda}} \boldsymbol{\Lambda} \boldsymbol{U}_{11} \boldsymbol{\Lambda} \boldsymbol{U}_{11}^\top) + D_1 \mathrm{trace}(\boldsymbol{\Lambda}^2 \boldsymbol{U}_{11}^\top), \tag{27}$$

*and $\boldsymbol{u}_{12}(t) = \boldsymbol{0}_d$, $\boldsymbol{u}_{21}(t) = \boldsymbol{0}_d$ for all $t \geq 0$, where*

$$D_1 = \begin{cases} n\sigma_w^2 + \frac{n(n+1)}{2}\sigma_e^2, & \lambda = \gamma = 1, \\ \frac{\lambda - \lambda^{n+1}}{1-\lambda}\sigma_w^2 + \frac{\lambda^{n+2}-(n+1)\lambda^2+n\lambda}{(1-\lambda)^2}\sigma_e^2, & \lambda \neq 1, \gamma = 1, \\ \frac{\gamma^{n+2}-\gamma^{2n+2}}{1-\gamma}\sigma_w^2 + \frac{\gamma-\gamma^{n+1}-\gamma^{n+2}+\gamma^{2n+2}}{(1-\gamma)^2(1+\gamma)}\sigma_e^2, & \lambda = 1, \gamma \neq 1, \\ \lambda^{2n+2}n\sigma_w^2 + \left( \frac{\lambda^2(1-\lambda^{2n})}{(1-\lambda^2)^2} - \frac{\lambda^{2n+2}}{1-\lambda^2}n \right)\sigma_e^2, & \lambda \neq 1, \gamma \neq 1, \lambda = \gamma, \\ \frac{\gamma-\gamma^{2n+1}}{\lambda-\gamma}\sigma_w^2 + \left( \frac{1}{1-\gamma^2}n - \frac{\gamma-\gamma^{2n+1}}{(\lambda-\gamma)(1-\gamma^2)} \right)\sigma_e^2, & \lambda \neq 1, \gamma \neq 1, \lambda = 1/\gamma, \\ \frac{\lambda^{n+1}\gamma^{n+2}-\lambda\gamma^{2n+2}}{\lambda-\gamma}\sigma_w^2 + \left( \frac{\lambda\gamma(1-\lambda^n\gamma^n)}{(1-\gamma^2)(1-\lambda\gamma)} - \frac{\lambda^{n+1}\gamma^{n+2}-\lambda\gamma^{2n+2}}{(\lambda-\gamma)(1-\gamma^2)} \right)\sigma_e^2, & \lambda \neq 1, \gamma \neq 1, \lambda \neq \gamma, \lambda \neq 1/\gamma. \end{cases}$$

*and $\widetilde{\boldsymbol{\Lambda}} = D_2(2\boldsymbol{\Lambda} + \mathrm{trace}(\boldsymbol{\Lambda})\boldsymbol{I}) + D_3\boldsymbol{\Lambda}$ with*

$$D_2 = \begin{cases} n\sigma_w^2 + \frac{(n+1)n}{2}\sigma_e^2, & \lambda = 1, \gamma = 1, \\ \frac{\lambda^2-\lambda^{2n+2}}{1-\lambda^2}\sigma_w^2 + \frac{(n+1)\lambda^2-\lambda^{2n+2}-n}{1-\lambda^2}\sigma_e^2, & \lambda \neq 1, \gamma = 1, \\ \frac{\gamma^2-\gamma^{2n+2}}{1-\gamma^2}\sigma_w^2 + \left( \frac{n\gamma^2}{1-\gamma^2} - \frac{\gamma^2-\gamma^{2n+2}}{(1-\gamma^2)^2} \right)\sigma_e^2, & \lambda = 1, \gamma \neq 1, \\ \lambda^{2n+2}n\sigma_w^2 - \left( \frac{n\lambda^{2n+2}}{1-\lambda^2} - \frac{\lambda^6-\lambda^{2n+4}}{(1-\lambda^2)^2} \right)\sigma_e^2, & \lambda \neq 1, \gamma \neq 1, \lambda = \gamma, \\ \frac{\lambda^{2n}-\gamma^{2n}}{\lambda^2-\gamma^2}\sigma_w^2 - \left( \frac{\lambda^{2n}-\gamma^{2n}}{(\lambda^2-\gamma^2)(1-\gamma^2)} - \frac{\lambda^2-\lambda^{2n}}{(1-\gamma^2)(1-\lambda^2)} \right)\sigma_e^2, & \lambda \neq 1, \gamma \neq 1, \lambda = 1/\gamma, \\ \frac{\gamma^2\lambda^{2n+2}-\lambda^2\gamma^{2n+2}}{\lambda^2-\gamma^2}\sigma_w^2 - \left( \frac{\gamma^2\lambda^{2n+2}-\lambda^2\gamma^{2n+2}}{(\lambda^2-\gamma^2)(1-\gamma^2)} - \frac{\gamma^2(\lambda^4-\lambda^{2n+2})}{(1-\gamma^2)(1-\lambda^2)} \right)\sigma_e^2, & \lambda \neq 1, \gamma \neq 1, \lambda \neq \gamma, \lambda \neq 1/\gamma, \end{cases}$$

*and*

$$D_3 = \begin{cases} n(n-1)\sigma_w^2 + \frac{(n-1)n(n+1)}{3}\sigma_e^2, & \lambda = 1, \gamma = 1, \\ \left( \frac{2(\lambda^{n+1}-\lambda^2)}{(1-\lambda)^2} - \frac{2(\lambda^{2n+1}-\lambda^3)}{(1-\lambda)^2(1+\lambda)} \right)\sigma_w^2 + \left( \frac{2(n\lambda^4-\lambda^{2n+4}-(n-1)\lambda^2)}{(1-\lambda)(1-\lambda^2)^2} + \frac{2(\lambda^n-n\lambda+n-1)}{(1-\lambda)^2(1+\lambda)\lambda^{n-2}} \right)\sigma_e^2, & \lambda \neq 1, \gamma = 1, \\ \left( 2\frac{\gamma^3-\gamma^{2n+1}}{(1-\gamma)^2(1+\gamma)} - 2\frac{\gamma^{n+2}-\gamma^{2n+1}}{(1-\gamma)^2} \right)\sigma_w^2 + \left( \frac{2}{\gamma^2-1}\left( \frac{\gamma^3-\gamma^{2n+1}}{(1-\gamma)^2(1+\gamma)} - \frac{\gamma^{n+2}-\gamma^{2n+1}}{(1-\gamma)^2} \right) \right. & \\ \left. - \frac{2\gamma^3}{(\gamma^2-1)(1-\gamma)}(n-1-\frac{\gamma^n-\gamma}{\gamma-1}) \right)\sigma_e^2, & \lambda = 1, \gamma \neq 1, \\ \lambda^{2n+2}n(n-1)\sigma_w^2 + \left( \frac{2n(\lambda^6-\lambda^{2n+4})}{(1-\lambda^2)^2} - \frac{2(\lambda^{2n+6}-n\lambda^8+(n-1)\lambda^6)}{(1-\lambda^2)^3} - \frac{\lambda^{2n+2}n(n-1)}{1-\lambda^2} \right)\sigma_e^2, & \lambda \neq 1, \gamma \neq 1, \lambda = \gamma, \\ \left( \frac{2\lambda^{2n}-2\gamma^{2n-4}}{\lambda(\lambda-\gamma)^2(\lambda+\gamma)} - \frac{2-2\gamma^{2n-2}}{\lambda-\gamma} \right)\sigma_w^2 + \left( \frac{2\lambda^{2n-2}-2}{(1-\gamma^2)(\lambda-\gamma)^2} - \frac{2(n-1)\gamma}{(1-\gamma^2)(\lambda-\gamma)} \right. & \\ \left. - \frac{2\lambda^{2n}-2\gamma^{2n-4}}{\lambda(\lambda-\gamma)^2(\lambda+\gamma)(1-\gamma^2)} + \frac{2-2\gamma^{2n-2}}{(1-\gamma^2)(\lambda-\gamma)} \right)\sigma_e^2, & \lambda \neq 1, \gamma \neq 1, \lambda = 1/\gamma, \\ \left( \frac{2\gamma^3\lambda^{2n+3}-2\lambda^5\gamma^{2n+1}}{\lambda(\lambda-\gamma)^2(\lambda+\gamma)} - \frac{2\gamma^{n+2}\lambda^{n+2}-2\gamma^{2n+1}\lambda^3}{\lambda-\gamma} \right)\sigma_w^2 + \left( \frac{2\gamma(\lambda^4-\lambda^{2n+2})}{(1-\gamma^2)(\lambda-\gamma)(1-\lambda^2)} \right. & \\ \left. - \frac{2(\lambda^3\gamma^2-\lambda^{n+2}\gamma^{n+1})}{(1-\gamma^2)(\lambda-\gamma)(1-\lambda\gamma)} - \frac{2\gamma^3\lambda^{2n+3}-2\lambda^5\gamma^{2n+1}}{\lambda(\lambda-\gamma)^2(\lambda+\gamma)(1-\gamma^2)} + \frac{2\gamma^{n+2}\lambda^{n+2}-2\gamma^{2n+1}\lambda^3}{(\lambda-\gamma)(1-\gamma^2)} \right)\sigma_e^2, & \lambda \neq 1, \gamma \neq 1, \lambda \neq \gamma, \lambda \neq 1/\gamma. \end{cases}$$

*Proof.* We first calculate the derivatives of $\boldsymbol{u}$ as following

$$\frac{\mathrm{d}\boldsymbol{u}(t)}{\mathrm{d}t} = -\nabla_{\boldsymbol{u}}L(\boldsymbol{u}) = -2\,\mathbb{E}[\langle\boldsymbol{H},\boldsymbol{u}\boldsymbol{u}^\top\rangle\boldsymbol{H}]\boldsymbol{u} + 2\,\mathbb{E}[\boldsymbol{w}_{n+1}^\top\boldsymbol{x}_{n+1}\boldsymbol{H}]\boldsymbol{u}. \tag{28}$$

Following the same analysis of (Zhang et al., 2024, Lemma 5.2), we can first obtain

$$2\,\mathbb{E}[\langle\boldsymbol{H},\boldsymbol{u}\boldsymbol{u}^\top\rangle\boldsymbol{H}\boldsymbol{u}]$$

$$= \frac{1}{2}\,\mathbb{E}\left[\mathrm{vec}(\boldsymbol{U})^\top\mathrm{vec}\left(\left(\sum_{i=1}^{n+1}\lambda^{n+1-i}\boldsymbol{z}_i\boldsymbol{z}_i^\top\right)\boldsymbol{U}\boldsymbol{X}\right)\mathrm{vec}\left(\left(\sum_{i=1}^{n+1}\lambda^{n+1-i}\boldsymbol{z}_i\boldsymbol{z}_i^\top\right)\boldsymbol{U}\boldsymbol{X}\right)\right]$$

$$= \frac{1}{2}\,\mathbb{E}\left[\sum_{i=1}^{d+1}\sum_{j=1}^{d+1}\boldsymbol{T}(i,j)\boldsymbol{U}(i,j)\,\mathrm{vec}(\boldsymbol{T})\right] \tag{29}$$

where $\boldsymbol{T} = \left(\sum_{i=1}^{n+1}\lambda^{n+1-i}\boldsymbol{z}_i\boldsymbol{z}_i^\top\right)\boldsymbol{U}\boldsymbol{X}$ and $\boldsymbol{H}$, $\boldsymbol{X}$ are defined in Lemma 2.

In addition, we can derive

$$2\,\mathbb{E}[\boldsymbol{w}_{n+1}^\top\boldsymbol{x}_{n+1}\boldsymbol{H}]\boldsymbol{u}$$

$$= \sum_{j=1}^d\mathbb{E}\left[(\boldsymbol{x}_{n+1}(j)\boldsymbol{X})\otimes\left(\boldsymbol{w}_{n+1}(j)\left(\sum_{i=1}^{n+1}\lambda^{n+1-i}\boldsymbol{z}_i\boldsymbol{z}_i^\top\right)\right)\right]\boldsymbol{u}$$

$$= \sum_{j=1}^d\left(\mathbb{E}\left[\boldsymbol{x}_{n+1}(j)\boldsymbol{X}\right]\otimes\mathbb{E}\left[\boldsymbol{w}_{n+1}(j)\left(\sum_{i=1}^{n+1}\lambda^{n+1-i}\boldsymbol{z}_i\boldsymbol{z}_i^\top\right)\right]\right)\boldsymbol{u}$$

$$= \sum_{j=1}^d\left(\mathbb{E}\left[\begin{bmatrix}\boldsymbol{0}_{d\times d} & \boldsymbol{x}_{n+1}(i)\boldsymbol{x}_{n+1} \\ \boldsymbol{x}_{n+1}(i)\boldsymbol{x}_{n+1}^\top & 0\end{bmatrix}\right]\right.$$

$$\left.\otimes\mathbb{E}\left[\begin{bmatrix}\sum_{i=1}^{n+1}\lambda^{n+1-i}\boldsymbol{w}_{n+1}(j)\boldsymbol{x}_i\boldsymbol{x}_i^\top & \sum_{i=1}^n\lambda^{n+1-i}\boldsymbol{w}_{n+1}(j)\boldsymbol{x}_i\boldsymbol{x}_i^\top\boldsymbol{w}_i \\ \sum_{i=1}^n\lambda^{n+1-i}\boldsymbol{w}_{n+1}(j)\boldsymbol{w}_i^\top\boldsymbol{x}_i\boldsymbol{x}_i^\top & \sum_{i=1}^n\lambda^{n+1-i}\boldsymbol{w}_{n+1}(j)\boldsymbol{w}_i^\top\boldsymbol{x}_i\boldsymbol{x}_i^\top\boldsymbol{w}_i\end{bmatrix}\right]\right)\boldsymbol{u}$$

$$= D_1\sum_{j=1}^d\left(\begin{bmatrix}\boldsymbol{0}_{d\times d} & \boldsymbol{\Lambda}(:,j) \\ \boldsymbol{\Lambda}^\top(:,j) & 0\end{bmatrix}\otimes\begin{bmatrix}\boldsymbol{0}_{d\times d} & \boldsymbol{\Lambda}(:,j) \\ \boldsymbol{\Lambda}^\top(:,j) & 0\end{bmatrix}\right)\boldsymbol{u}, \tag{30}$$

where the second equation follows the fact that $\{\boldsymbol{w}_i\}$ are independent of $\{\boldsymbol{x}_i\}$, and the last line follows $\boldsymbol{w}_0\overset{\text{i.i.d.}}{\sim}\mathcal{N}(\boldsymbol{0},\sigma_w^2\mathbf{I})$, $\boldsymbol{e}_i\overset{\text{i.i.d.}}{\sim}\mathcal{N}(\boldsymbol{0},\sigma_e^2\mathbf{I})$, $\boldsymbol{x}_i\overset{\text{i.i.d.}}{\sim}\mathcal{N}(\boldsymbol{0},\boldsymbol{\Lambda})$ and they are mutually independent. In addition, to compute $D_1$, we apply Lemma 1 to evaluate the cross-covariance $\mathbb{E}[\boldsymbol{w}_{n+1}\boldsymbol{w}_i^\top] =$
$$\begin{cases}(\sigma_w^2 + i\sigma_e^2)\mathbf{I}, & \gamma = 1 \\ (\gamma^{n+1+i}\sigma_w^2 + \frac{\gamma^{n+1+i}-\gamma^{1+n-i}}{\gamma^2-1}\sigma_e^2)\mathbf{I}, & \gamma \neq 1\end{cases}$$ and then get

$$D_1 = \begin{cases}\sum_{i=1}^n\lambda^{n+1-i}(\sigma_w^2 + i\sigma_e^2), & \gamma = 1 \\ \sum_{i=1}^n\lambda^{n+1-i}(\gamma^{n+1+i}\sigma_w^2 + \frac{\gamma^{n+1+i}-\gamma^{1+n-i}}{\gamma^2-1}\sigma_e^2), & \gamma \neq 1\end{cases}$$

$$= \begin{cases}n\sigma_w^2 + \frac{n(n+1)}{2}\sigma_e^2, & \lambda = \gamma = 1, \\ \frac{\lambda-\lambda^{n+1}}{1-\lambda}\sigma_w^2 + \frac{\lambda^{n+2}-(n+1)\lambda^2+n\lambda}{(1-\lambda)^2}\sigma_e^2, & \lambda \neq 1, \gamma = 1, \\ \frac{\gamma^{n+2}-\gamma^{2n+2}}{1-\gamma}\sigma_w^2 + \frac{\gamma-\gamma^{n+1}-\gamma^{n+2}+\gamma^{2n+2}}{(1-\gamma)^2(1+\gamma)}\sigma_e^2, & \lambda = 1, \gamma \neq 1, \\ \lambda^{2n+2}n\sigma_w^2 + \left(\frac{\lambda^2(1-\lambda^{2n})}{(1-\lambda^2)^2} - \frac{\lambda^{2n+2}}{1-\lambda^2}n\right)\sigma_e^2, & \lambda \neq 1, \gamma \neq 1, \lambda = \gamma, \\ \frac{\gamma-\gamma^{2n+1}}{\lambda-\gamma}\sigma_w^2 + \left(\frac{1}{1-\gamma^2}n - \frac{\gamma-\gamma^{2n+1}}{(\lambda-\gamma)(1-\gamma^2)}\right)\sigma_e^2, & \lambda \neq 1, \gamma \neq 1, \lambda = 1/\gamma, \\ \frac{\lambda^{n+1}\gamma^{n+2}-\lambda\gamma^{2n+2}}{\lambda-\gamma}\sigma_w^2 + \left(\frac{\lambda\gamma(1-\lambda^n\gamma^n)}{(1-\gamma^2)(1-\lambda\gamma)} - \frac{\lambda^{n+1}\gamma^{n+2}-\lambda\gamma^{2n+2}}{(\lambda-\gamma)(1-\gamma^2)}\right)\sigma_e^2, & \lambda \neq 1, \gamma \neq 1, \lambda \neq \gamma, \lambda \neq 1/\gamma.\end{cases} \tag{31}$$

Combing (29) and (30), (28) can be rewritten as

$$\frac{\mathrm{d}\boldsymbol{u}(t)}{\mathrm{d}t} = -\frac{1}{2}\,\mathbb{E}\left[\sum_{i=1}^{d+1}\sum_{j=1}^{d+1}\boldsymbol{T}(i,j)\boldsymbol{U}(i,j)\,\mathrm{vec}(\boldsymbol{T})\right] + D_1\sum_{j=1}^d\left(\begin{bmatrix}\boldsymbol{0}_{d\times d} & \boldsymbol{\Lambda}(:,j) \\ \boldsymbol{\Lambda}^\top(:,j) & 0\end{bmatrix}\otimes\begin{bmatrix}\boldsymbol{0}_{d\times d} & \boldsymbol{\Lambda}(:,j) \\ \boldsymbol{\Lambda}^\top(:,j) & 0\end{bmatrix}\right)\boldsymbol{u}. \tag{32}$$

**Dynamics of $U$**   Under the Assumption 1, following the same analysis of (Zhang et al., 2024, Lemma 5.2), we can guarantee $u_{12}(t) = 0_d \in \mathbb{R}^d$ and $u_{21}(t) = 0_d \in \mathbb{R}^d$ for all time $t \geq 0$.

Now, we will respectively analyze dynamics of $U_{11}$ and $u_{-1}$ when $u_{12}(t) = 0_d \in \mathbb{R}^d$ and $u_{21}(t) = 0_d \in \mathbb{R}^d$.

**Dynamics of $U_{11}$**   First, for $k, l \in [d]$, we use $\frac{\mathrm{d}u(t)}{\mathrm{d}t}((l-1)(d+1)+k) = \frac{\mathrm{d}U(t)}{\mathrm{d}t}(k,l)$ and then expand (29) as

$$\frac{1}{2} \mathbb{E}\left[ \sum_{i=1}^{d+1} \sum_{j=1}^{d+1} T(i,j)U(i,j)T(k,l) \right]$$

$$= \frac{1}{2} \mathbb{E}\left[ \sum_{i=1}^{d} \sum_{j=1}^{d} T(i,j)U(i,j)T(k,l) \right] + \frac{1}{2}\mathbb{E}[T(d+1,d+1)u_{-1}T(k,l)]. \tag{33}$$

For the term $T(i,j)U(i,j)T(k,l)$, we have

$$\mathbb{E}[T(i,j)U(i,j)T(k,l)]$$

$$= U(i,j)u_{-1}^2 \mathbb{E}\left[ \left( \sum_{a=1}^{n} \lambda^{n+1-a} x_a(i) x_a^\top w_a x_{n+1}(j) \right) \left( \sum_{b=1}^{n} \lambda^{n+1-b} x_{n+1}(l) w_b^\top x_b x_b(k) \right) \right]$$

$$= U(i,j)u_{-1}^2 \Lambda(j,l) \mathbb{E}\left[ \left( \sum_{a=1}^{n} \lambda^{n+1-a} x_a(i) x_a^\top w_a \right) \left( \sum_{b=1}^{n} \lambda^{n+1-b} w_b^\top x_b x_b(k) \right) \right]$$

$$= \begin{cases} U(i,j)u_{-1}^2 \Lambda(j,l) \mathbb{E}\left[ \sum_{a=1}^{n} \sum_{b=1}^{n} \lambda^{2n+2-a-b}(\sigma_w^2 + \min\{a,b\}\sigma_e^2) x_a(i) x_a^\top x_b x_b(k) \right], & \gamma = 1, \\ U(i,j)u_{-1}^2 \Lambda(j,l) \mathbb{E}\left[ \sum_{a=1}^{n} \sum_{b=1}^{n} \lambda^{2n+2-a-b}(\gamma^{a+b}\sigma_w^2 + \frac{\gamma^{a+b}-\gamma^{|a-b|}}{\gamma^2-1}\sigma_e^2) x_a(i) x_a^\top x_b x_b(k) \right], & \gamma \neq 1, \end{cases} \tag{34}$$

where the last line follows Lemma 1. For the summation, we have

$$\frac{1}{2} \mathbb{E}\left[ \sum_{i=1}^{d} \sum_{j=1}^{d} T(i,j)U(i,j)T(k,l) \right]$$

$$= \begin{cases} \frac{1}{2}u_{-1}^2 \mathbb{E}\left[ \sum_{a=1}^{n} \sum_{b=1}^{n} \lambda^{2n+2-a-b}(\sigma_w^2 + \min\{a,b\}\sigma_e^2) x_b(k) x_b^\top x_a x_a^\top \right] U_{11}\Lambda(:,l), & \gamma = 1, \\ \frac{1}{2}u_{-1}^2 \mathbb{E}\left[ \sum_{a=1}^{n} \sum_{b=1}^{n} \lambda^{2n+2-a-b}(\gamma^{a+b}\sigma_w^2 + \frac{\gamma^{a+b}-\gamma^{|a-b|}}{\gamma^2-1}\sigma_e^2) x_b(k) x_b^\top x_a x_a^\top \right] U_{11}\Lambda(:,l), & \gamma \neq 1. \end{cases} \tag{35}$$

For the term $T(d+1,d+1)u_{-1}T(k,l)$, we get

$$\frac{1}{2} \mathbb{E}[T(d+1,d+1)u_{-1}T(k,l)]$$

$$= \frac{1}{2}u_{-1}^2 \mathbb{E}\left[ \left( \sum_{a=1}^{n} \lambda^{n+1-a} w_a^\top x_a x_a^\top U_{11} x_{n+1} \right) \left( \sum_{b=1}^{n} \lambda^{n+1-b} x_{n+1}(l) w_b^\top x_b x_b(k) \right) \right]$$

$$= \frac{1}{2}u_{-1}^2 \mathbb{E}\left[ \left( \sum_{a=1}^{n} \sum_{b=1}^{n} \lambda^{2n+2-a-b} x_b(k) x_b^\top w_b w_a^\top x_a x_a^\top \right) U_{11} x_{n+1} x_{n+1}(l) \right]$$

$$= \begin{cases} \frac{1}{2}u_{-1}^2 \mathbb{E}\left[ \sum_{a=1}^{n} \sum_{b=1}^{n} \lambda^{2n+2-a-b}(\sigma_w^2 + \min\{a,b\}\sigma_e^2) x_b(k) x_b^\top x_a x_a^\top \right] U_{11}\Lambda(:,l), & \gamma = 1, \\ \frac{1}{2}u_{-1}^2 \mathbb{E}\left[ \sum_{a=1}^{n} \sum_{b=1}^{n} \lambda^{2n+2-a-b}(\gamma^{a+b}\sigma_w^2 + \frac{\gamma^{a+b}-\gamma^{|a-b|}}{\gamma^2-1}\sigma_e^2) x_b(k) x_b^\top x_a x_a^\top \right] U_{11}\Lambda(:,l), & \gamma \neq 1. \end{cases} \tag{36}$$

Combing (35) and (36), we have

$$\frac{1}{2} \mathbb{E}\left[ \sum_{i=1}^{d+1} \sum_{j=1}^{d+1} T(i,j)U(i,j)T(k,l) \right]$$

$$= \begin{cases} u_{-1}^2 \mathbb{E}\left[ \sum_{a=1}^{n} \sum_{b=1}^{n} \lambda^{2n+2-a-b}(\sigma_w^2 + \min\{a,b\}\sigma_e^2) x_b(k) x_b^\top x_a x_a^\top \right] U_{11}\Lambda(:,l), & \gamma = 1, \\ u_{-1}^2 \mathbb{E}\left[ \sum_{a=1}^{n} \sum_{b=1}^{n} \lambda^{2n+2-a-b}(\gamma^{a+b}\sigma_w^2 + \frac{\gamma^{a+b}-\gamma^{|a-b|}}{\gamma^2-1}\sigma_e^2) x_b(k) x_b^\top x_a x_a^\top \right] U_{11}\Lambda(:,l), & \gamma \neq 1. \end{cases} \tag{37}$$

In addition, using (30) and following the analysis of (Zhang et al., 2024, Lemma 5.2), the $((l-1)(d+1)+k)$-th element of the second term in (32) can be computed as $D_1\mathbf{\Lambda}^\top(:,k)\mathbf{\Lambda}(:,l)u_{-1}$. Furthermore, we can obtain

$$
\frac{\mathrm{d}\boldsymbol{U}_{11}(t)}{\mathrm{d}t}
$$
$$
= D_1 u_{-1}\mathbf{\Lambda}^2 - \begin{cases} u_{-1}^2\,\mathbb{E}\big[\sum_{a=1}^n\sum_{b=1}^n \lambda^{2n+2-a-b}(\sigma_w^2+\min\{a,b\}\sigma_e^2)\boldsymbol{x}_b\boldsymbol{x}_b^\top\boldsymbol{x}_a\boldsymbol{x}_a^\top\big]\boldsymbol{U}_{11}\mathbf{\Lambda}, & \gamma=1, \\ u_{-1}^2\,\mathbb{E}\big[\sum_{a=1}^n\sum_{b=1}^n \lambda^{2n+2-a-b}(\gamma^{a+b}\sigma_w^2+\frac{\gamma^{a+b}-\gamma^{|a-b|}}{\gamma^2-1}\sigma_e^2)\boldsymbol{x}_b\boldsymbol{x}_b^\top\boldsymbol{x}_a\boldsymbol{x}_a^\top\big]\boldsymbol{U}_{11}\mathbf{\Lambda}, & \gamma\neq1. \end{cases}
$$

Notice that based on (Sayed, 2011, Lemma A.2), $\mathbb{E}\big[\boldsymbol{x}_b\boldsymbol{x}_b^\top\boldsymbol{x}_a\boldsymbol{x}_a^\top\big]=\begin{cases}\mathbf{\Lambda}^2, & a\neq b \\ 2\mathbf{\Lambda}^2+\mathrm{trace}(\mathbf{\Lambda})\mathbf{\Lambda}, & a=b\end{cases}.$
We can further derive

$$
\begin{cases} \mathbb{E}\big[\sum_{a=1}^n\sum_{b=1}^n \lambda^{2n+2-a-b}(\sigma_w^2+\min\{a,b\}\sigma_e^2)\boldsymbol{x}_b\boldsymbol{x}_b^\top\boldsymbol{x}_a\boldsymbol{x}_a^\top\big], & \gamma=1 \\ \mathbb{E}\big[\sum_{a=1}^n\sum_{b=1}^n \lambda^{2n+2-a-b}(\gamma^{a+b}\sigma_w^2+\frac{\gamma^{a+b}-\gamma^{|a-b|}}{\gamma^2-1}\sigma_e^2)\boldsymbol{x}_b\boldsymbol{x}_b^\top\boldsymbol{x}_a\boldsymbol{x}_a^\top\big], & \gamma\neq1 \end{cases}
$$
$$
= \begin{cases} \sum_{a=1}^n \lambda^{2n+2-2a}(\sigma_w^2+a\sigma_e^2)(2\mathbf{\Lambda}^2+\mathrm{trace}(\mathbf{\Lambda})\mathbf{\Lambda})+2\sum_{b=1}^{n-1}\sum_{a=b+1}^n \lambda^{2n+2-a-b}(\sigma_w^2+b\sigma_e^2)\mathbf{\Lambda}^2, & \gamma=1 \\ \sum_{a=1}^n \lambda^{2n+2-2a}(\gamma^{2a}\sigma_w^2+\frac{\gamma^{2a}-1}{\gamma^2-1}\sigma_e^2)(2\mathbf{\Lambda}^2+\mathrm{trace}(\mathbf{\Lambda})\mathbf{\Lambda}) \\ \qquad +2\sum_{b=1}^{n-1}\sum_{a=b+1}^n \lambda^{2n+2-a-b}(\gamma^{a+b}\sigma_w^2+\frac{\gamma^{a+b}-\gamma^{a-b}}{\gamma^2-1}\sigma_e^2)\mathbf{\Lambda}^2, & \gamma\neq1 \end{cases}
$$

$$
= \begin{cases} (n\sigma_w^2+\frac{(n+1)n}{2}\sigma_e^2)(2\mathbf{\Lambda}^2+\mathrm{trace}(\mathbf{\Lambda})\mathbf{\Lambda})+(n(n-1)\sigma_w^2+\frac{(n-1)n(n+1)}{3}\sigma_e^2)\mathbf{\Lambda}^2, & \lambda=1,\gamma=1, \\[4pt] (\frac{\lambda^2-\lambda^{2n+2}}{1-\lambda^2}\sigma_w^2+\frac{(n+1)\lambda^2-\lambda^{2n+2}-n}{1-\lambda^2}\sigma_e^2)(2\mathbf{\Lambda}^2+\mathrm{trace}(\mathbf{\Lambda})\mathbf{\Lambda}) \\ \quad+((\frac{2(\lambda^{n+1}-\lambda^2)}{(1-\lambda)^2}-\frac{2(\lambda^{2n+1}-\lambda^3)}{(1-\lambda)^2(1+\lambda)})\sigma_w^2+(\frac{2(n\lambda^4-\lambda^{2n+4}-(n-1)\lambda^2)}{(1-\lambda)(1-\lambda^2)^2}+\frac{2(\lambda^n-n\lambda+n-1)}{(1-\lambda)^2(1+\lambda)\lambda^{n-2}})\sigma_e^2)\mathbf{\Lambda}^2, & \lambda\neq1,\gamma=1, \\[4pt] (\frac{\gamma^2-\gamma^{2n+2}}{1-\gamma^2}\sigma_w^2+(\frac{n}{1-\gamma^2}-\frac{\gamma^2-\gamma^{2n+2}}{(1-\gamma^2)^2})\sigma_e^2)(2\mathbf{\Lambda}^2+\mathrm{trace}(\mathbf{\Lambda})\mathbf{\Lambda}) \\ \quad+((2\frac{\gamma^3-\gamma^{2n+1}}{(1-\gamma)^2(1+\gamma)}-2\frac{\gamma^{n+2}-\gamma^{2n+1}}{(1-\gamma)^2})\sigma_w^2+(\frac{2}{\gamma^2-1}(\frac{\gamma^3-\gamma^{2n+1}}{(1-\gamma)^2(1+\gamma)}-\frac{\gamma^{n+2}-\gamma^{2n+1}}{(1-\gamma)^2}) \\ \qquad-\frac{2\gamma}{(\gamma^2-1)(1-\gamma)}(n-1-\frac{\gamma^n-\gamma}{\gamma-1}))\sigma_e^2)\mathbf{\Lambda}^2, & \lambda=1,\gamma\neq1, \\[4pt] (\lambda^{2n+2}n\sigma_w^2-(\frac{n\lambda^{2n+2}}{1-\lambda^2}-\frac{\lambda^4-\lambda^{2n+2}}{(1-\lambda^2)^2})\sigma_e^2)(2\mathbf{\Lambda}^2+\mathrm{trace}(\mathbf{\Lambda})\mathbf{\Lambda}) \\ \quad+(\lambda^{2n+2}n(n-1)\sigma_w^2+(\frac{2n(\lambda^4-\lambda^{2n+2})}{(1-\lambda^2)^2}-\frac{2(\lambda^{2n+4}-n\lambda^6+(n-1)\lambda^4)}{(1-\lambda^2)^3}-\frac{\lambda^{2n+2}n(n-1)}{1-\lambda^2})\sigma_e^2)\mathbf{\Lambda}^2, & \lambda\neq1,\gamma\neq1,\lambda=\gamma, \\[4pt] (\frac{\lambda^{2n}-\gamma^{2n}}{\lambda^2-\gamma^2}\sigma_w^2-(\frac{\lambda^{2n}-\gamma^{2n}}{(\lambda^2-\gamma^2)(1-\gamma^2)}-\frac{1-\lambda^{2n-2}}{(1-\gamma^2)(1-\lambda^2)})\sigma_e^2)(2\mathbf{\Lambda}^2+\mathrm{trace}(\mathbf{\Lambda})\mathbf{\Lambda}) \\ \quad+((\frac{2\lambda^{2n}-2\gamma^{2n-4}}{\lambda(\lambda-\gamma)^2(\lambda+\gamma)}-\frac{2-2\gamma^{2n-2}}{\lambda-\gamma})\sigma_w^2+(\frac{2\lambda^{2n}-2\lambda^2}{(1-\gamma^2)(\lambda-\gamma)^2}-\frac{2(n-1)\gamma\lambda^2}{(1-\gamma^2)(\lambda-\gamma)} \\ \qquad-\frac{2\lambda^{2n}-2\gamma^{2n-4}}{\lambda(\lambda-\gamma)^2(\lambda+\gamma)(1-\gamma^2)}+\frac{2-2\gamma^{2n-2}}{(1-\gamma^2)(\lambda-\gamma)})\sigma_e^2)\mathbf{\Lambda}^2, & \lambda\neq1,\gamma\neq1,\lambda=1/\gamma, \\[4pt] (\frac{\gamma^2\lambda^{2n+2}-\lambda^2\gamma^{2n+2}}{\lambda^2-\gamma^2}\sigma_w^2-(\frac{\gamma^2\lambda^{2n+2}-\lambda^2\gamma^{2n+2}}{(\lambda^2-\gamma^2)(1-\gamma^2)}-\frac{\lambda^2-\lambda^{2n+2}}{(1-\gamma^2)(1-\lambda^2)})\sigma_e^2)(2\mathbf{\Lambda}^2+\mathrm{trace}(\mathbf{\Lambda})\mathbf{\Lambda}) \\ \quad+((\frac{2\gamma^3\lambda^{2n+3}-2\lambda^5\gamma^{2n+1}}{\lambda(\lambda-\gamma)^2(\lambda+\gamma)}-\frac{2\gamma^{n+2}\lambda^{n+2}-2\gamma^{2n+1}\lambda^3}{\lambda-\gamma})\sigma_w^2+(\frac{2\gamma^{-1}(\lambda^4-\lambda^{2n+2})}{(1-\gamma^2)(\lambda-\gamma)(1-\lambda^2)} \\ \qquad-\frac{2(\lambda^3-\lambda^{n+2}\gamma^{n-1})}{(1-\gamma^2)(\lambda-\gamma)(1-\lambda\gamma)}-\frac{2\gamma^3\lambda^{2n+3}-2\lambda^5\gamma^{2n+1}}{\lambda(\lambda-\gamma)^2(\lambda+\gamma)(1-\gamma^2)}+\frac{2\gamma^{n+2}\lambda^{n+2}-2\gamma^{2n+1}\lambda^3}{(\lambda-\gamma)(1-\gamma^2)})\sigma_e^2)\mathbf{\Lambda}^2, & \lambda\neq1,\gamma\neq1,\lambda\neq\gamma,\lambda\neq1/\gamma. \end{cases}
\tag{38}
$$

Now, we can obtain the dynamics of $\boldsymbol{U}_{11}$ as following

$$
\frac{\mathrm{d}\boldsymbol{U}_{11}(t)}{\mathrm{d}t} = -\boldsymbol{u}_{-1}^2\widetilde{\mathbf{\Lambda}}\mathbf{\Lambda}\boldsymbol{U}_{11}\mathbf{\Lambda}+D_1 u_{-1}\mathbf{\Lambda}^2,
\tag{39}
$$

where $\widetilde{\mathbf{\Lambda}}=D_2(2\mathbf{\Lambda}+\mathrm{trace}(\mathbf{\Lambda})\mathbf{I})+D_3\mathbf{\Lambda}$ with

$$
D_2 = \begin{cases} n\sigma_w^2+\frac{(n+1)n}{2}\sigma_e^2, & \lambda=1,\gamma=1, \\[4pt] \frac{\lambda^2-\lambda^{2n+2}}{1-\lambda^2}\sigma_w^2+\frac{(n+1)\lambda^2-\lambda^{2n+2}-n}{1-\lambda^2}\sigma_e^2, & \lambda\neq1,\gamma=1, \\[4pt] \frac{\gamma^2-\gamma^{2n+2}}{1-\gamma^2}\sigma_w^2+(\frac{n}{1-\gamma^2}-\frac{\gamma^2-\gamma^{2n+2}}{(1-\gamma^2)^2})\sigma_e^2, & \lambda=1,\gamma\neq1, \\[4pt] \lambda^{2n+2}n\sigma_w^2-(\frac{n\lambda^{2n+2}}{1-\lambda^2}-\frac{\lambda^4-\lambda^{2n+2}}{(1-\lambda^2)^2})\sigma_e^2, & \lambda\neq1,\gamma\neq1,\lambda=\gamma, \\[4pt] \frac{\lambda^{2n}-\gamma^{2n}}{\lambda^2-\gamma^2}\sigma_w^2-(\frac{\lambda^{2n}-\gamma^{2n}}{(\lambda^2-\gamma^2)(1-\gamma^2)}-\frac{1-\lambda^{2n-2}}{(1-\gamma^2)(1-\lambda^2)})\sigma_e^2, & \lambda\neq1,\gamma\neq1,\lambda=1/\gamma, \\[4pt] \frac{\gamma^2\lambda^{2n+2}-\lambda^2\gamma^{2n+2}}{\lambda^2-\gamma^2}\sigma_w^2-(\frac{\gamma^2\lambda^{2n+2}-\lambda^2\gamma^{2n+2}}{(\lambda^2-\gamma^2)(1-\gamma^2)}-\frac{\lambda^2-\lambda^{2n+2}}{(1-\gamma^2)(1-\lambda^2)})\sigma_e^2, & \lambda\neq1,\gamma\neq1,\lambda\neq\gamma,\lambda\neq1/\gamma, \end{cases}
$$

and

$$D_3 = \begin{cases} n(n-1)\sigma_w^2 + \frac{(n-1)n(n+1)}{3}\sigma_e^2, & \lambda=1, \gamma=1, \\ (\frac{2(\lambda^{n+1}-\lambda^2)}{(1-\lambda)^2} - \frac{2(\lambda^{2n+1}-\lambda^3)}{(1-\lambda)^2(1+\lambda)})\sigma_w^2 + (\frac{2(n\lambda^4-\lambda^{2n+4}-(n-1)\lambda^2)}{(1-\lambda)(1-\lambda^2)^2} + \frac{2(\lambda^n-n\lambda+n-1)}{(1-\lambda)^2(1+\lambda)\lambda^{n-2}})\sigma_e^2, & \lambda \neq 1, \gamma=1, \\ (2\frac{\gamma^3-\gamma^{2n+1}}{(1-\gamma)^2(1+\gamma)} - 2\frac{\gamma^{n+2}-\gamma^{2n+1}}{(1-\gamma)^2})\sigma_w^2 + (\frac{2}{\gamma^2-1}(\frac{\gamma^3-\gamma^{2n+1}}{(1-\gamma)^2(1+\gamma)} - \frac{\gamma^{n+2}-\gamma^{2n+1}}{(1-\gamma)^2}) \\ \quad - \frac{2\gamma}{(\gamma^2-1)(1-\gamma)}(n-1-\frac{\gamma^n-\gamma}{\gamma-1}))\sigma_e^2, & \lambda=1, \gamma\neq 1, \\ \lambda^{2n+2}n(n-1)\sigma_w^2 + (\frac{2n(\lambda^4-\lambda^{2n+2})}{(1-\lambda^2)^2} - \frac{2(\lambda^{2n+4}-n\lambda^6+(n-1)\lambda^4)}{(1-\lambda^2)^3} - \frac{\lambda^{2n+2}n(n-1)}{1-\lambda^2})\sigma_e^2, & \lambda \neq 1, \gamma \neq 1, \lambda=\gamma, \\ (\frac{2\lambda^{2n}-2\gamma^{2n-4}}{\lambda(\lambda-\gamma)^2(\lambda+\gamma)} - \frac{2-2\gamma^{2n-2}}{\lambda-\gamma})\sigma_w^2 + (\frac{2\lambda^{2n}-2\lambda^2}{(1-\gamma^2)(\lambda-\gamma)^2} - \frac{2(n-1)\gamma\lambda^2}{(1-\gamma^2)(\lambda-\gamma)} \\ \quad - \frac{2\lambda^{2n}-2\gamma^{2n-4}}{\lambda(\lambda-\gamma)^2(\lambda+\gamma)(1-\gamma^2)} + \frac{2-2\gamma^{2n-2}}{(1-\gamma^2)(\lambda-\gamma)})\sigma_e^2, & \lambda \neq 1, \gamma \neq 1, \lambda=1/\gamma, \\ (\frac{2\gamma^3\lambda^{2n+3}-2\lambda^5\gamma^{2n+1}}{\lambda(\lambda-\gamma)^2(\lambda+\gamma)} - \frac{2\gamma^{n+2}\lambda^{n+2}-2\gamma^{2n+1}\lambda^3}{\lambda-\gamma})\sigma_w^2 + (\frac{2\gamma^{-1}(\lambda^4-\lambda^{2n+2})}{(1-\gamma^2)(\lambda-\gamma)(1-\lambda^2)} \\ \quad - \frac{2(\lambda^3-\lambda^{n+2}\gamma^{n-1})}{(1-\gamma^2)(\lambda-\gamma)(1-\lambda\gamma)} - \frac{2\gamma^3\lambda^{2n+3}-2\lambda^5\gamma^{2n+1}}{\lambda(\lambda-\gamma)^2(\lambda+\gamma)(1-\gamma^2)} + \frac{2\gamma^{n+2}\lambda^{n+2}-2\gamma^{2n+1}\lambda^3}{(\lambda-\gamma)(1-\gamma^2)})\sigma_e^2, & \lambda \neq 1, \gamma \neq 1, \lambda \neq \gamma, \lambda \neq 1/\gamma. \end{cases}$$

**Dynamics of $u_{-1}$**  Next, similar with (33), we expand (29) at $(d+1)^2$-th element:

$$\frac{1}{2}\,\mathbb{E}\left[\sum_{i=1}^{d+1}\sum_{j=1}^{d+1}\boldsymbol{T}(i,j)\boldsymbol{U}(i,j)\boldsymbol{T}(d+1,d+1)\right]$$

$$= \frac{1}{2}\,\mathbb{E}\left[\sum_{i=1}^{d}\sum_{j=1}^{d}\boldsymbol{T}(i,j)\boldsymbol{U}(i,j)\boldsymbol{T}(d+1,d+1)\right] + \frac{1}{2}\,\mathbb{E}[\boldsymbol{T}(d+1,d+1)u_{-1}\boldsymbol{T}(d+1,d+1)]$$

$$= \frac{u_{-1}}{2}\sum_{i=1}^{n}\sum_{j=1}^{n}\boldsymbol{U}(i,j)\,\mathbb{E}\left[\left(\sum_{a=1}^{n}\lambda^{n+1-a}\boldsymbol{x}_a(i)\boldsymbol{x}_a^\top\boldsymbol{w}_a\boldsymbol{x}_{n+1}(j)\right)\left(\sum_{b=1}^{n}\lambda^{n+1-b}\boldsymbol{w}_b^\top\boldsymbol{x}_b\boldsymbol{x}_b^\top\boldsymbol{U}_{11}\boldsymbol{x}_{n+1}\right)\right]$$

$$+ \frac{u_{-1}}{2}\,\mathbb{E}\left[\left(\sum_{a=1}^{n}\lambda^{n+1-a}\boldsymbol{w}_a^\top\boldsymbol{x}_a\boldsymbol{x}_a^\top\boldsymbol{U}_{11}\boldsymbol{x}_{n+1}\right)\left(\sum_{b=1}^{n}\lambda^{n+1-b}\boldsymbol{x}_{n+1}^\top\boldsymbol{U}_{11}^\top\boldsymbol{x}_b\boldsymbol{x}_b^\top\boldsymbol{w}_b\right)\right]$$

$$= \frac{u_{-1}}{2}\,\mathrm{trace}\left(\sum_{i=1}^{n}\sum_{j=1}^{n}\boldsymbol{\Lambda}(:,j)\boldsymbol{U}(i,j)\,\mathbb{E}\left[\sum_{a=1}^{n}\sum_{b=1}^{n}\lambda^{2n+2-a-b}\boldsymbol{x}_a(i)\boldsymbol{x}_a^\top\boldsymbol{w}_a\boldsymbol{w}_b^\top\boldsymbol{x}_b\boldsymbol{x}_b^\top\right]\boldsymbol{U}_{11}\right)$$

$$+ \frac{u_{-1}}{2}\,\mathbb{E}\left[\mathrm{trace}\left(\left(\sum_{a=1}^{n}\lambda^{n+1-a}\boldsymbol{w}_a^\top\boldsymbol{x}_a\boldsymbol{x}_a^\top\right)\boldsymbol{U}_{11}\boldsymbol{\Lambda}\boldsymbol{U}_{11}^\top\left(\sum_{b=1}^{n}\lambda^{n+1-b}\boldsymbol{x}_b\boldsymbol{x}_b^\top\boldsymbol{w}_b\right)\right)\right]$$

$$= \frac{u_{-1}}{2}\,\mathrm{trace}\left(\boldsymbol{\Lambda}\boldsymbol{U}_{11}^\top\,\mathbb{E}\left[\sum_{a=1}^{n}\sum_{b=1}^{n}\lambda^{2n+2-a-b}\boldsymbol{x}_a\boldsymbol{x}_a^\top\boldsymbol{w}_a\boldsymbol{w}_b^\top\boldsymbol{x}_b\boldsymbol{x}_b^\top\right]\boldsymbol{U}_{11}\right)$$

$$+ \frac{u_{-1}}{2}\,\mathbb{E}\left[\mathrm{trace}\left(\sum_{a=1}^{n}\sum_{b=1}^{n}\lambda^{2n+2-a-b}\boldsymbol{x}_b\boldsymbol{x}_b^\top\boldsymbol{w}_b\boldsymbol{w}_a^\top\boldsymbol{x}_a\boldsymbol{x}_a^\top\boldsymbol{U}_{11}\boldsymbol{\Lambda}\boldsymbol{U}_{11}^\top\right)\right]$$

$$= u_{-1}\,\mathrm{trace}(\widetilde{\boldsymbol{\Lambda}}\boldsymbol{\Lambda}\boldsymbol{U}_{11}\boldsymbol{\Lambda}\boldsymbol{U}_{11}^\top), \tag{40}$$

where the last line follows (36) and (39).

In addition, according to (Zhang et al., 2024, Lemma 5.2), the last element of the second term in (32) can be represented as $D_1\,\mathrm{trace}(\boldsymbol{\Lambda}^2\boldsymbol{U}_{11}^\top)$. Hence, we have

$$\frac{\mathrm{d}u_{-1}(t)}{\mathrm{d}t} = -\mathrm{trace}(u_{-1}\widetilde{\boldsymbol{\Lambda}}\boldsymbol{\Lambda}\boldsymbol{U}_{11}\boldsymbol{\Lambda}\boldsymbol{U}_{11}^\top) + D_1\,\mathrm{trace}(\boldsymbol{\Lambda}^2\boldsymbol{U}_{11}^\top). \tag{41}$$

$\square$

Based on the same analysis of (Zhang et al., 2024, Lemma A.1), we have

**Lemma 4.** *Consider the gradient flow of $L$ in (12) with respect to $\boldsymbol{u}$, initialized according to Assumption 1. This is equivalent to performing gradient flow with respect to $\boldsymbol{U}_{11}$ and $u_{-1}$ on the same loss function*

$$\widetilde{L}(\boldsymbol{U}_{11}, u_{-1}) = \frac{u_{-1}^2}{2}\,\mathrm{trace}(\widetilde{\boldsymbol{\Lambda}}\boldsymbol{\Lambda}\boldsymbol{U}_{11}\boldsymbol{\Lambda}\boldsymbol{U}_{11}^\top) - D_1 u_{-1}\,\mathrm{trace}(\boldsymbol{\Lambda}^2\boldsymbol{U}_{11}^\top), \tag{42}$$

*where $\widetilde{\boldsymbol{\Lambda}}$ and $D_1$ are defined in Lemma 3.*

Note that $\widetilde{L}(\boldsymbol{U}_{11}, u_{-1})$ differs from $L(\boldsymbol{u})$ by a constant and can therefore take negative values. We can further derive

**Corollary 1.** *The loss function $\widetilde{L}(\boldsymbol{U}_{11}, u_{-1})$ in Lemma 4 satisfies*

$$\min_{\substack{\boldsymbol{U}_{11} \in \mathbb{R}^{d \times d}, \\ u_{-1} \in \mathbb{R}}} \widetilde{L}(\boldsymbol{U}_{11}, u_{-1}) = -\frac{D_1^2}{2} \operatorname{trace}(\boldsymbol{\Lambda}^2 \widetilde{\boldsymbol{\Lambda}}^{-1}), \tag{43}$$

*and*

$$\widetilde{L}(\boldsymbol{U}_{11}, u_{-1}) - \min_{\substack{\boldsymbol{U}_{11} \in \mathbb{R}^{d \times d}, \\ u_{-1} \in \mathbb{R}}} \widetilde{L}(\boldsymbol{U}_{11}, u_{-1}) = \frac{1}{2} \|\widetilde{\boldsymbol{\Lambda}}^{\frac{1}{2}} (u_{-1} \boldsymbol{\Lambda}^{\frac{1}{2}} \boldsymbol{U}_{11} \boldsymbol{\Lambda}^{\frac{1}{2}} - D_1 \boldsymbol{\Lambda} \widetilde{\boldsymbol{\Lambda}}^{-1}) \|_F^2. \tag{44}$$

*Furthermore, $L$ in (12) satisfies that*

$$L(\boldsymbol{U}_{11}, u_{-1}) - \min_{\substack{\boldsymbol{U}_{11} \in \mathbb{R}^{d \times d}, \\ u_{-1} \in \mathbb{R}}} L(\boldsymbol{U}_{11}, u_{-1}) = \frac{1}{2} \|\widetilde{\boldsymbol{\Lambda}}^{\frac{1}{2}} (u_{-1} \boldsymbol{\Lambda}^{\frac{1}{2}} \boldsymbol{U}_{11} \boldsymbol{\Lambda}^{\frac{1}{2}} - D_1 \boldsymbol{\Lambda} \widetilde{\boldsymbol{\Lambda}}^{-1}) \|_F^2. \tag{45}$$

*and the global minimum $(\boldsymbol{U}_{11}, u_{-1})$ of $L$ in (12), when initialized according to Assumption 1, satisfies*

$$u_{-1} \boldsymbol{U}_{11} = D_1 \widetilde{\boldsymbol{\Lambda}}^{-1}. \tag{46}$$

*Proof.* First, we testify $\frac{1}{2} \operatorname{trace}(\widetilde{\boldsymbol{\Lambda}}(u_{-1} \boldsymbol{\Lambda}^{\frac{1}{2}} \boldsymbol{U}_{11} \boldsymbol{\Lambda}^{\frac{1}{2}} - D_1 \boldsymbol{\Lambda} \widetilde{\boldsymbol{\Lambda}}^{-1})(u_{-1} \boldsymbol{\Lambda}^{\frac{1}{2}} \boldsymbol{U}_{11} \boldsymbol{\Lambda}^{\frac{1}{2}} - D_1 \boldsymbol{\Lambda} \widetilde{\boldsymbol{\Lambda}}^{-1})^\top) - \frac{D_1^2}{2} \operatorname{trace}(\boldsymbol{\Lambda}^2 \widetilde{\boldsymbol{\Lambda}}^{-1})$ is equivalent to $\widetilde{L}(\boldsymbol{U}_{11}, u_{-1})$. Specifically, we have

$$\frac{1}{2} \operatorname{trace} \left( \widetilde{\boldsymbol{\Lambda}} (u_{-1} \boldsymbol{\Lambda}^{\frac{1}{2}} \boldsymbol{U}_{11} \boldsymbol{\Lambda}^{\frac{1}{2}} - D_1 \boldsymbol{\Lambda} \widetilde{\boldsymbol{\Lambda}}^{-1})(u_{-1} \boldsymbol{\Lambda}^{\frac{1}{2}} \boldsymbol{U}_{11} \boldsymbol{\Lambda}^{\frac{1}{2}} - D_1 \boldsymbol{\Lambda} \widetilde{\boldsymbol{\Lambda}}^{-1})^\top \right)$$

$$- \frac{D_1^2}{2} \operatorname{trace}(\boldsymbol{\Lambda}^2 \widetilde{\boldsymbol{\Lambda}}^{-1})$$

$$= \frac{1}{2} \operatorname{trace} \left( \widetilde{\boldsymbol{\Lambda}} (u_{-1}^2 \boldsymbol{\Lambda}^{\frac{1}{2}} \boldsymbol{U}_{11} \boldsymbol{\Lambda} \boldsymbol{U}_{11}^\top \boldsymbol{\Lambda}^{\frac{1}{2}} - D_1 u_{-1} \boldsymbol{\Lambda} \widetilde{\boldsymbol{\Lambda}}^{-1} \boldsymbol{\Lambda}^{\frac{1}{2}} \boldsymbol{U}_{11} \boldsymbol{\Lambda}^{\frac{1}{2}} \right.$$

$$\left. - D_1 u_1 \boldsymbol{\Lambda}^{\frac{1}{2}} \boldsymbol{U}_{11} \boldsymbol{\Lambda}^{\frac{3}{2}} \widetilde{\boldsymbol{\Lambda}}^{-1} + D_1^2 \widetilde{\boldsymbol{\Lambda}}^{-2} \boldsymbol{\Lambda}^2) \right) - \frac{D_1^2}{2} \operatorname{trace}(\boldsymbol{\Lambda}^2 \widetilde{\boldsymbol{\Lambda}}^{-1})$$

$$= \frac{u_{-1}^2}{2} \operatorname{trace}(\widetilde{\boldsymbol{\Lambda}} \boldsymbol{\Lambda} \boldsymbol{U}_{11} \boldsymbol{\Lambda} \boldsymbol{U}_{11}^\top) - D_1 u_{-1} \operatorname{trace}(\boldsymbol{\Lambda}^2 \boldsymbol{U}_{11}^\top)$$

$$= \widetilde{L}(\boldsymbol{U}_{11}, u_{-1}), \tag{47}$$

where the second equation uses the fact that $\widetilde{\boldsymbol{\Lambda}}$ and $\boldsymbol{\Lambda}$ commute.

Notice that $\widetilde{\boldsymbol{\Lambda}}$ and $(u_{-1} \boldsymbol{\Lambda}^{\frac{1}{2}} \boldsymbol{U}_{11} \boldsymbol{\Lambda}^{\frac{1}{2}} - D_1 \boldsymbol{\Lambda} \widetilde{\boldsymbol{\Lambda}}^{-1})(u_{-1} \boldsymbol{\Lambda}^{\frac{1}{2}} \boldsymbol{U}_{11} \boldsymbol{\Lambda}^{\frac{1}{2}} - D_1 \boldsymbol{\Lambda} \widetilde{\boldsymbol{\Lambda}}^{-1})^\top$ are positive semidefinite matrices, we have $\frac{1}{2} \operatorname{trace}(\widetilde{\boldsymbol{\Lambda}}(u_{-1} \boldsymbol{\Lambda}^{\frac{1}{2}} \boldsymbol{U}_{11} \boldsymbol{\Lambda}^{\frac{1}{2}} - D_1 \boldsymbol{\Lambda} \widetilde{\boldsymbol{\Lambda}}^{-1})(u_{-1} \boldsymbol{\Lambda}^{\frac{1}{2}} \boldsymbol{U}_{11} \boldsymbol{\Lambda}^{\frac{1}{2}} - D_1 \boldsymbol{\Lambda} \widetilde{\boldsymbol{\Lambda}}^{-1})^\top) \geq 0$. Hence, we have $\min_{\substack{\boldsymbol{U}_{11} \in \mathbb{R}^{d \times d}, \\ u_{-1} \in \mathbb{R}}} \widetilde{L}(\boldsymbol{U}_{11}, u_{-1}) = -\frac{D_1^2}{2} \operatorname{trace}(\boldsymbol{\Lambda}^2 \widetilde{\boldsymbol{\Lambda}}^{-1})$. In addition, (44) follows $\operatorname{trace}(\boldsymbol{A}^\top \boldsymbol{A}) = \|\boldsymbol{A}\|_F^2$ for any matrix $\boldsymbol{A}$. Since $L(\boldsymbol{U}_{11}, u_{-1}) = \widetilde{L}(\boldsymbol{U}_{11}, u_{-1}) + C$ for some constant $C$, condition (45) is satisfied.

To attain the global minimum characterized by (43), it is necessary that $u_{-1} \boldsymbol{\Lambda}^{\frac{1}{2}} \boldsymbol{U}_{11} \boldsymbol{\Lambda}^{\frac{1}{2}} - D_1 \boldsymbol{\Lambda} \widetilde{\boldsymbol{\Lambda}}^{-1} = \boldsymbol{0}$. Using the identity $\boldsymbol{\Lambda} \widetilde{\boldsymbol{\Lambda}}^{-1} = \boldsymbol{\Lambda}^{\frac{1}{2}} \widetilde{\boldsymbol{\Lambda}}^{-1} \boldsymbol{\Lambda}^{\frac{1}{2}}$, this condition reduces to $u_{-1} \boldsymbol{U}_{11} = D_1 \widetilde{\boldsymbol{\Lambda}}^{-1}$.

□

Following the same analysis of (Zhang et al., 2024, Lemma A.3, Lemma A.4 and Lemma A.5), we can directly obtain

**Lemma 5.** *Consider the gradient flow of $L$ in (12) with respect to $\boldsymbol{u}$, initialized according to Assumption 1. For any $t \geq 0$, it holds that*

$$u_{-1}^2(t) = \operatorname{trace}(\boldsymbol{U}_{11}(t) \boldsymbol{U}_{11}^\top(t)). \tag{48}$$

**Lemma 6.** *Consider the gradient flow of $L$ in* (12) *with respect to $\boldsymbol{u}$, initialized according to Assumption 1. If the initial scale satisfies*

$$0 < \sigma < \sqrt{\frac{2D_1}{\sqrt{d}\|\widetilde{\boldsymbol{\Lambda}}\|}}, \tag{49}$$

*then, for any $t \geq 0$, it holds that*

$$u_{-1}(t) \geq \sqrt{\frac{\sigma^2}{2D_1\sqrt{d}\|\boldsymbol{\Lambda}\|^2}\|\boldsymbol{\Lambda}\boldsymbol{\Theta}\|_F^2(2D_1 - \sqrt{d}\sigma^2\|\widetilde{\boldsymbol{\Lambda}}\|)} > 0. \tag{50}$$

## B    PROOF OF THEOREM 1

The following theorem is established under a more general setting with $\lambda, \gamma > 0$.

**Theorem 4.** *Consider gradient flow over the population loss in* (12). *Assume that the initial task weight $\boldsymbol{w}_0 \overset{i.i.d.}{\sim} \mathcal{N}(\mathbf{0}, \sigma_w^2\mathbf{I})$, noises $\boldsymbol{e}_i \overset{i.i.d.}{\sim} \mathcal{N}(\mathbf{0}, \sigma_e^2\mathbf{I})$ and inputs $\boldsymbol{x}_i \overset{i.i.d.}{\sim} \mathcal{N}(\mathbf{0}, \boldsymbol{\Lambda})$. Suppose the initialization satisfies Assumption 1 with initialization scale $\sigma > 0$ satisfying $\sigma < \sqrt{\frac{2D_1}{\sqrt{d}\|\widetilde{\boldsymbol{\Lambda}}\|}}$ where*

$$D_1 = \begin{cases} n\sigma_w^2 + \frac{n(n+1)}{2}\sigma_e^2, & \lambda = \gamma = 1, \\ \frac{\lambda-\lambda^{n+1}}{1-\lambda}\sigma_w^2 + \frac{\lambda^{n+2}-(n+1)\lambda^2+n\lambda}{(1-\lambda)^2}\sigma_e^2, & \lambda \neq 1, \gamma = 1, \\ \frac{\gamma^{n+2}-\gamma^{2n+2}}{1-\gamma}\sigma_w^2 + \frac{\gamma-\gamma^{n+1}-\gamma^{n+2}+\gamma^{2n+2}}{(1-\gamma)^2(1+\gamma)}\sigma_e^2, & \lambda = 1, \gamma \neq 1, \\ \lambda^{2n+2}n\sigma_w^2 + \left(\frac{\lambda^2(1-\lambda^{2n})}{(1-\lambda^2)^2} - \frac{\lambda^{2n+2}}{1-\lambda^2}n\right)\sigma_e^2, & \lambda \neq 1, \gamma \neq 1, \lambda = \gamma, \\ \frac{\gamma-\gamma^{2n+1}}{\lambda-\gamma}\sigma_w^2 + \left(\frac{1}{1-\gamma^2}n - \frac{\gamma-\gamma^{2n+1}}{(\lambda-\gamma)(1-\gamma^2)}\right)\sigma_e^2, & \lambda \neq 1, \gamma \neq 1, \lambda = 1/\gamma, \\ \frac{\lambda^{n+1}\gamma^{n+2}-\lambda\gamma^{2n+2}}{\lambda-\gamma}\sigma_w^2 + \left(\frac{\lambda\gamma(1-\lambda^n\gamma^n)}{(1-\gamma^2)(1-\lambda\gamma)} - \frac{\lambda^{n+1}\gamma^{n+2}-\lambda\gamma^{2n+2}}{(\lambda-\gamma)(1-\gamma^2)}\right)\sigma_e^2, & \lambda \neq 1, \gamma \neq 1, \lambda \neq \gamma, \lambda \neq 1/\gamma, \end{cases}$$

*and $\widetilde{\boldsymbol{\Lambda}} = D_2(2\boldsymbol{\Lambda} + \mathrm{trace}(\boldsymbol{\Lambda})\mathbf{I}) + D_3\boldsymbol{\Lambda}$ with*

$$D_2 = \begin{cases} n\sigma_w^2 + \frac{(n+1)n}{2}\sigma_e^2, & \lambda = 1, \gamma = 1, \\ \frac{\lambda^2-\lambda^{2n+2}}{1-\lambda^2}\sigma_w^2 + \frac{(n+1)\lambda^2-\lambda^{2n+2}-n}{1-\lambda^2}\sigma_e^2, & \lambda \neq 1, \gamma = 1, \\ \frac{\gamma^2-\gamma^{2n+2}}{1-\gamma^2}\sigma_w^2 + \left(\frac{n}{1-\gamma^2} - \frac{\gamma^2-\gamma^{2n+2}}{(1-\gamma^2)^2}\right)\sigma_e^2, & \lambda = 1, \gamma \neq 1, \\ \lambda^{2n+2}n\sigma_w^2 - \left(\frac{n\lambda^{2n+2}}{1-\lambda^2} - \frac{\lambda^4-\lambda^{2n+2}}{(1-\lambda^2)^2}\right)\sigma_e^2, & \lambda \neq 1, \gamma \neq 1, \lambda = \gamma, \\ \frac{\lambda^{2n}-\gamma^{2n}}{\lambda^2-\gamma^2}\sigma_w^2 - \left(\frac{\lambda^{2n}-\gamma^{2n}}{(\lambda^2-\gamma^2)(1-\gamma^2)} - \frac{1-\lambda^{2n-2}}{(1-\gamma^2)(1-\lambda^2)}\right)\sigma_e^2, & \lambda \neq 1, \gamma \neq 1, \lambda = 1/\gamma, \\ \frac{\gamma^2\lambda^{2n+2}-\lambda^2\gamma^{2n+2}}{\lambda^2-\gamma^2}\sigma_w^2 - \left(\frac{\gamma^2\lambda^{2n+2}-\lambda^2\gamma^{2n+2}}{(\lambda^2-\gamma^2)(1-\gamma^2)} - \frac{\lambda^2-\lambda^{2n+2}}{(1-\gamma^2)(1-\lambda^2)}\right)\sigma_e^2, & \lambda \neq 1, \gamma \neq 1, \lambda \neq \gamma, \lambda \neq 1/\gamma, \end{cases}$$

*and*

$$D_3 = \begin{cases} n(n-1)\sigma_w^2 + \frac{(n-1)n(n+1)}{3}\sigma_e^2, & \lambda = 1, \gamma = 1, \\ \left(\frac{2(\lambda^{n+1}-\lambda^2)}{(1-\lambda)^2} - \frac{2(\lambda^{2n+1}-\lambda^3)}{(1-\lambda)^2(1+\lambda)}\right)\sigma_w^2 + \left(\frac{2(n\lambda^4-\lambda^{2n+4}-(n-1)\lambda^2)}{(1-\lambda)(1-\lambda^2)^2} + \frac{2(\lambda^n-n\lambda+n-1)}{(1-\lambda)^2(1+\lambda)\lambda^{n-2}}\right)\sigma_e^2, & \lambda \neq 1, \gamma = 1, \\ \left(2\frac{\gamma^3-\gamma^{2n+1}}{(1-\gamma)^2(1+\gamma)} - 2\frac{\gamma^{n+2}-\gamma^{2n+1}}{(1-\gamma)^2}\right)\sigma_w^2 + \left(\frac{2}{\gamma^2-1}\left(\frac{\gamma^3-\gamma^{2n+1}}{(1-\gamma)^2(1+\gamma)} - \frac{\gamma^{n+2}-\gamma^{2n+1}}{(1-\gamma)^2}\right)\right. \\ \quad \left. - \frac{2\gamma}{(\gamma^2-1)(1-\gamma)}(n-1-\frac{\gamma^n-\gamma}{\gamma-1})\right)\sigma_e^2, & \lambda = 1, \gamma \neq 1, \\ \lambda^{2n+2}n(n-1)\sigma_w^2 + \left(\frac{2n(\lambda^4-\lambda^{2n+2})}{(1-\lambda^2)^2} - \frac{2(\lambda^{2n+4}-n\lambda^6+(n-1)\lambda^4)}{(1-\lambda^2)^3} - \frac{\lambda^{2n+2}n(n-1)}{1-\lambda^2}\right)\sigma_e^2, & \lambda \neq 1, \gamma \neq 1, \lambda = \gamma, \\ \left(\frac{2\lambda^{2n}-2\gamma^{2n-4}}{\lambda(\lambda-\gamma)^2(\lambda+\gamma)} - \frac{2-2\gamma^{2n-2}}{\lambda-\gamma}\right)\sigma_w^2 + \left(\frac{2\lambda^{2n}-2\lambda^2}{(1-\gamma^2)(\lambda-\gamma)^2} - \frac{2(n-1)\gamma\lambda^2}{(1-\gamma^2)(\lambda-\gamma)}\right. \\ \quad \left. - \frac{2\lambda^{2n}-2\gamma^{2n-4}}{\lambda(\lambda-\gamma)^2(\lambda+\gamma)(1-\gamma^2)} + \frac{2-2\gamma^{2n-2}}{(1-\gamma^2)(\lambda-\gamma)}\right)\sigma_e^2, & \lambda \neq 1, \gamma \neq 1, \lambda = 1/\gamma, \\ \left(\frac{2\gamma^3\lambda^{2n+3}-2\lambda^5\gamma^{2n+1}}{\lambda(\lambda-\gamma)^2(\lambda+\gamma)} - \frac{2\gamma^{n+2}\lambda^{n+2}-2\gamma^{2n+1}\lambda^3}{\lambda-\gamma}\right)\sigma_w^2 + \left(\frac{2\gamma^{-1}(\lambda^4-\lambda^{2n+2})}{(1-\gamma^2)(\lambda-\gamma)(1-\lambda^2)}\right. \\ \quad \left. - \frac{2(\lambda^3-\lambda^{n+2}\gamma^{n-1})}{(1-\gamma^2)(\lambda-\gamma)(1-\lambda\gamma)} - \frac{2\gamma^3\lambda^{2n+3}-2\lambda^5\gamma^{2n+1}}{\lambda(\lambda-\gamma)^2(\lambda+\gamma)(1-\gamma^2)} + \frac{2\gamma^{n+2}\lambda^{n+2}-2\gamma^{2n+1}\lambda^3}{(\lambda-\gamma)(1-\gamma^2)}\right)\sigma_e^2, & \lambda \neq 1, \gamma \neq 1, \lambda \neq \gamma, \lambda \neq 1/\gamma. \end{cases}$$

*Then gradient flow converges to a global minimum of the population loss* (12). *Moreover, $\boldsymbol{W}_{KQ}(0)$ and $\boldsymbol{W}_V(0)$ respectively converge to*

$$\lim_{t\to\infty}\boldsymbol{W}_V(t) = \sqrt{D_1\|\widetilde{\boldsymbol{\Lambda}}^{-1}\|_F}\begin{bmatrix}\mathbf{0}_{d\times d} & \mathbf{0}_d \\ \mathbf{0}_d^\top & 1\end{bmatrix} \quad \text{and} \quad \lim_{t\to\infty}\boldsymbol{W}_{KQ}(t) = \sqrt{D_1\|\widetilde{\boldsymbol{\Lambda}}^{-1}\|_F^{-1}}\begin{bmatrix}\widetilde{\boldsymbol{\Lambda}}^{-1} & \mathbf{0}_d \\ \mathbf{0}_d^\top & 0\end{bmatrix}. \tag{51}$$

*Proof.* We begin by deriving a lower bound on the squared gradient norm of the loss function

$$
\begin{aligned}
&\|\nabla L(\boldsymbol{U}_{11}(t), u_{-1}(t))\|_2^2 \\
&= \left\|\frac{\partial L(\boldsymbol{U}_{11}(t), u_{-1}(t))}{\partial \boldsymbol{U}_{11}(t)}\right\|_F^2 + \left|\frac{\partial L(\boldsymbol{U}_{11}(t), u_{-1}(t))}{\partial u_{-1}(t)}\right|^2 \geq \left\|\frac{\partial L(\boldsymbol{U}_{11}(t), u_{-1}(t))}{\partial \boldsymbol{U}_{11}(t)}\right\|_F^2 \\
&= \|u_{-1}^2 \widetilde{\boldsymbol{\Lambda}}\boldsymbol{\Lambda}\boldsymbol{U}_{11}\boldsymbol{\Lambda} - D_1 u_{-1}\boldsymbol{\Lambda}^2\|_F^2 \\
&= u_{-1}^2 \|\widetilde{\boldsymbol{\Lambda}}\boldsymbol{\Lambda}^{\frac{1}{2}}(u_{-1}\boldsymbol{\Lambda}^{\frac{1}{2}}\boldsymbol{U}_{11}\boldsymbol{\Lambda}^{\frac{1}{2}} - \boldsymbol{\Lambda}\widetilde{\boldsymbol{\Lambda}}^{-1})\boldsymbol{\Lambda}^{\frac{1}{2}}\|_F^2 \\
&\geq \frac{\sigma^2}{2D_1\sqrt{d}\|\boldsymbol{\Lambda}\|^2}\|\boldsymbol{\Lambda}\boldsymbol{\Theta}\|_F^2(2D_1 - \sqrt{d}\sigma^2\|\widetilde{\boldsymbol{\Lambda}}\|)\|\widetilde{\boldsymbol{\Lambda}}\boldsymbol{\Lambda}^{\frac{1}{2}}(u_{-1}\boldsymbol{\Lambda}^{\frac{1}{2}}\boldsymbol{U}_{11}\boldsymbol{\Lambda}^{\frac{1}{2}} - \boldsymbol{\Lambda}\widetilde{\boldsymbol{\Lambda}}^{-1})\boldsymbol{\Lambda}^{\frac{1}{2}}\|_F^2, \quad (52)
\end{aligned}
$$

where the third equation uses that $\widetilde{\boldsymbol{\Lambda}}$ and $\boldsymbol{\Lambda}$ commute and the last line follows Lemma 6.

In addition, based on Corollary 1, we have

$$
\begin{aligned}
&L(\boldsymbol{U}_{11}(t), u_{-1}(t)) - \min_{\substack{\boldsymbol{U}_{11}\in\mathbb{R}^{d\times d}, \\ u_{-1}\in\mathbb{R}}} L(\boldsymbol{U}_{11}, u_{-1}) \\
&= \frac{1}{2}\|\widetilde{\boldsymbol{\Lambda}}^{\frac{1}{2}}(u_{-1}\boldsymbol{\Lambda}^{\frac{1}{2}}\boldsymbol{U}_{11}\boldsymbol{\Lambda}^{\frac{1}{2}} - D_1\boldsymbol{\Lambda}\widetilde{\boldsymbol{\Lambda}}^{-1})\|_F^2 \\
&\leq \frac{1}{2}\|\widetilde{\boldsymbol{\Lambda}}\boldsymbol{\Lambda}^{\frac{1}{2}}(u_{-1}\boldsymbol{\Lambda}^{\frac{1}{2}}\boldsymbol{U}_{11}\boldsymbol{\Lambda}^{\frac{1}{2}} - D_1\boldsymbol{\Lambda}\widetilde{\boldsymbol{\Lambda}}^{-1})\boldsymbol{\Lambda}^{\frac{1}{2}}\|_F^2\|\widetilde{\boldsymbol{\Lambda}}^{-\frac{1}{2}}\boldsymbol{\Lambda}^{-\frac{1}{2}}\|_F^2\|\boldsymbol{\Lambda}^{-\frac{1}{2}}\|_F^2 \\
&= \frac{1}{2}\|\widetilde{\boldsymbol{\Lambda}}\boldsymbol{\Lambda}^{\frac{1}{2}}(u_{-1}\boldsymbol{\Lambda}^{\frac{1}{2}}\boldsymbol{U}_{11}\boldsymbol{\Lambda}^{\frac{1}{2}} - D_1\boldsymbol{\Lambda}\widetilde{\boldsymbol{\Lambda}}^{-1})\boldsymbol{\Lambda}^{\frac{1}{2}}\|_F^2 \operatorname{trace}(\widetilde{\boldsymbol{\Lambda}}^{-1}\boldsymbol{\Lambda}^{-1}) \operatorname{trace}(\boldsymbol{\Lambda}^{-1}), \quad (53)
\end{aligned}
$$

where the first inequality uses that $\widetilde{\boldsymbol{\Lambda}}$ and $\boldsymbol{\Lambda}$ commute.

Combing (52) and (53), we can get Polyak–Łojasiewicz (PL) inequality as follows:

$$
\begin{aligned}
&\|\nabla L(\boldsymbol{U}_{11}(t), u_{-1}(t))\|_2^2 \\
&\geq \frac{\sigma^2\|\boldsymbol{\Lambda}\boldsymbol{\Theta}\|_F^2(2D_1 - \sqrt{d}\sigma^2\|\widetilde{\boldsymbol{\Lambda}}\|)}{D_1\sqrt{d}\|\boldsymbol{\Lambda}\|^2 \operatorname{trace}(\widetilde{\boldsymbol{\Lambda}}^{-1}\boldsymbol{\Lambda}^{-1}) \operatorname{trace}(\boldsymbol{\Lambda}^{-1})}\Big(L(\boldsymbol{U}_{11}(t), u_{-1}(t)) - \min_{\substack{\boldsymbol{U}_{11}\in\mathbb{R}^{d\times d}, \\ u_{-1}\in\mathbb{R}}} L(\boldsymbol{U}_{11}, u_{-1})\Big) \\
&:= \alpha\Big(L(\boldsymbol{U}_{11}(t), u_{-1}(t)) - \min_{\substack{\boldsymbol{U}_{11}\in\mathbb{R}^{d\times d}, \\ u_{-1}\in\mathbb{R}}} L(\boldsymbol{U}_{11}, u_{-1})\Big). \quad (54)
\end{aligned}
$$

From the dynamics of gradient flow and the PL condition, we have

$$
\begin{aligned}
&\frac{\mathrm{d}}{\mathrm{d}t}\Big(L(\boldsymbol{U}_{11}(t), u_{-1}(t)) - \min_{\substack{\boldsymbol{U}_{11}\in\mathbb{R}^{d\times d}, \\ u_{-1}\in\mathbb{R}}} L(\boldsymbol{U}_{11}, u_{-1})\Big) \\
&= \left\langle \frac{\mathrm{d}\boldsymbol{U}_{11}(t)}{\mathrm{d}t}, \frac{\partial L(\boldsymbol{U}_{11}(t), u_{-1}(t))}{\partial \boldsymbol{U}_{11}(t)}\right\rangle + \left\langle \frac{\mathrm{d}u_{-1}(t)}{\mathrm{d}t}, \frac{\partial L(\boldsymbol{U}_{11}(t), u_{-1}(t))}{\partial u_{-1}(t)}\right\rangle \\
&= -\left\|\frac{\partial L(\boldsymbol{U}_{11}(t), u_{-1}(t))}{\partial \boldsymbol{U}_{11}(t)}\right\|_F^2 - \left|\frac{\partial L(\boldsymbol{U}_{11}(t), u_{-1}(t))}{\partial u_{-1}(t)}\right|^2 \\
&\leq -\alpha\Big(L(\boldsymbol{U}_{11}(t), u_{-1}(t)) - \min_{\substack{\boldsymbol{U}_{11}\in\mathbb{R}^{d\times d}, \\ u_{-1}\in\mathbb{R}}} L(\boldsymbol{U}_{11}, u_{-1})\Big). \quad (55)
\end{aligned}
$$

When $t \to \infty$, we have

$$
\begin{aligned}
0 &\leq L(\boldsymbol{U}_{11}(t), u_{-1}(t)) - \min_{\substack{\boldsymbol{U}_{11}\in\mathbb{R}^{d\times d}, \\ u_{-1}\in\mathbb{R}}} L(\boldsymbol{U}_{11}, u_{-1}) \\
&\leq e^{-\alpha t}\Big(L(\boldsymbol{U}_{11}(0), u_{-1}(0)) - \min_{\substack{\boldsymbol{U}_{11}\in\mathbb{R}^{d\times d}, \\ u_{-1}\in\mathbb{R}}} L(\boldsymbol{U}_{11}, u_{-1})\Big) \to 0, \quad (56)
\end{aligned}
$$

which implies $\lim_{t\to\infty} L(\boldsymbol{U}_{11}(t), u_{-1}(t)) - \min_{\substack{\boldsymbol{U}_{11}\in\mathbb{R}^{d\times d}, \\ u_{-1}\in\mathbb{R}}} L(\boldsymbol{U}_{11}, u_{-1}) = 0.$

According to Corollary 1, Lemma 5 and Lemma 6, we have $u_{-1}(t)\boldsymbol{U}_{11}(t) = D_1\widetilde{\boldsymbol{\Lambda}}^{-1}$, $u_{-1}(t) = \|\boldsymbol{U}_{11}(t)\|_F$ and further obtain

$$\lim_{t\to\infty} \boldsymbol{U}_{11}(t) = \sqrt{D_1\|\widetilde{\boldsymbol{\Lambda}}^{-1}\|_F^{-1}}\widetilde{\boldsymbol{\Lambda}}^{-1}, \quad \lim_{t\to\infty} u_{-1}(t) = \sqrt{D_1\|\widetilde{\boldsymbol{\Lambda}}^{-1}\|_F}. \tag{57}$$

This completes the proof.

$\square$

## C    PROOF OF THEOREM 2

Similar with Theorem 4, the following theorem is also established under a more general setting with $\lambda, \gamma > 0$.

**Theorem 5.** *(Training error) Assuming the conditions in Theorem 4 hold, the recovery error between* (15) *and* (2) *is*

$$\mathbb{E}[(\widehat{y}_{n+1} - y_{n+1})^2] = D_1^2\, \mathrm{trace}\left(D_2(\boldsymbol{\Lambda}\, \mathrm{trace}(\widetilde{\boldsymbol{\Lambda}}^{-1}\boldsymbol{\Lambda}\widetilde{\boldsymbol{\Lambda}}^{-1}\boldsymbol{\Lambda}) + 2\boldsymbol{\Lambda}\widetilde{\boldsymbol{\Lambda}}^{-1}\boldsymbol{\Lambda}\widetilde{\boldsymbol{\Lambda}}^{-1}\boldsymbol{\Lambda}\right)$$
$$+ D_3\boldsymbol{\Lambda}\widetilde{\boldsymbol{\Lambda}}^{-1}\boldsymbol{\Lambda}\widetilde{\boldsymbol{\Lambda}}^{-1}\boldsymbol{\Lambda}) + D_4\, \mathrm{trace}(\boldsymbol{\Lambda}) - 2D_1^2\, \mathrm{trace}(\boldsymbol{\Lambda}\widetilde{\boldsymbol{\Lambda}}^{-1}\boldsymbol{\Lambda}), \tag{58}$$

*where* $D_4 = \begin{cases} \sigma_w^2 + (n+1)\sigma_e^2, & \gamma = 1 \\ \gamma^{2n+2}\sigma_w^2 + \frac{1-\gamma^{2n+2}}{1-\gamma^2}\sigma_e^2, & \gamma \neq 1 \end{cases}$.

*Proof.* To establish the result, we expand $\mathbb{E}[(\widehat{y}_{n+1} - y_{n+1})^2]$ as

$$\mathbb{E}[(\widehat{y}_{n+1} - y_{n+1})^2]$$
$$= \mathbb{E}\left[\left(D_1\left(\sum_{i=1}^n \lambda^{n+1-i}\boldsymbol{w}_i^\top \boldsymbol{x}_i\boldsymbol{x}_i^\top\right)\widetilde{\boldsymbol{\Lambda}}^{-1}\boldsymbol{x}_{n+1} - \boldsymbol{w}_{n+1}^\top\boldsymbol{x}_{n+1}\right)^2\right]$$
$$= D_1^2\, \mathbb{E}\left[\left(\sum_{i=1}^n \lambda^{n+1-i}\boldsymbol{w}_i^\top \boldsymbol{x}_i\boldsymbol{x}_i^\top\right)\widetilde{\boldsymbol{\Lambda}}^{-1}\boldsymbol{x}_{n+1}\boldsymbol{x}_{n+1}^\top\widetilde{\boldsymbol{\Lambda}}^{-1}\left(\sum_{j=1}^n \lambda^{n+1-j}\boldsymbol{x}_j\boldsymbol{x}_j^\top\boldsymbol{w}_j\right)\right]$$
$$+ \mathbb{E}[\boldsymbol{w}_{n+1}^\top\boldsymbol{x}_{n+1}\boldsymbol{x}_{n+1}^\top\boldsymbol{w}_{n+1}] - 2D_1\, \mathbb{E}\left[\left(\sum_{i=1}^n \lambda^{n+1-i}\boldsymbol{w}_i^\top \boldsymbol{x}_i\boldsymbol{x}_i^\top\right)\widetilde{\boldsymbol{\Lambda}}^{-1}\boldsymbol{x}_{n+1}\boldsymbol{x}_{n+1}^\top\boldsymbol{w}_{n+1}\right]. \tag{59}$$

For the first term of the last equation in (59), we have

$$\mathbb{E}\left[\left(\sum_{i=1}^n \lambda^{n+1-i}\boldsymbol{w}_i^\top \boldsymbol{x}_i\boldsymbol{x}_i^\top\right)\widetilde{\boldsymbol{\Lambda}}^{-1}\boldsymbol{x}_{n+1}\boldsymbol{x}_{n+1}^\top\widetilde{\boldsymbol{\Lambda}}^{-1}\left(\sum_{j=1}^n \lambda^{n+1-j}\boldsymbol{x}_j\boldsymbol{x}_j^\top\boldsymbol{w}_j\right)\right]$$
$$= \mathbb{E}\left[\mathrm{trace}\left(\sum_{i=1}^n\sum_{j=1}^n \lambda^{2n+2-i-j}\boldsymbol{x}_i\boldsymbol{x}_i^\top\widetilde{\boldsymbol{\Lambda}}^{-1}\boldsymbol{\Lambda}\widetilde{\boldsymbol{\Lambda}}^{-1}\boldsymbol{x}_j\boldsymbol{x}_j^\top\boldsymbol{w}_j\boldsymbol{w}_i^\top\right)\right]$$
$$= D_2\, \mathrm{trace}(\boldsymbol{\Lambda}\, \mathrm{trace}(\widetilde{\boldsymbol{\Lambda}}^{-1}\boldsymbol{\Lambda}\widetilde{\boldsymbol{\Lambda}}^{-1}\boldsymbol{\Lambda}) + 2\boldsymbol{\Lambda}\widetilde{\boldsymbol{\Lambda}}^{-1}\boldsymbol{\Lambda}\widetilde{\boldsymbol{\Lambda}}^{-1}\boldsymbol{\Lambda}) + D_3\, \mathrm{trace}(\boldsymbol{\Lambda}\widetilde{\boldsymbol{\Lambda}}^{-1}\boldsymbol{\Lambda}\widetilde{\boldsymbol{\Lambda}}^{-1}\boldsymbol{\Lambda}), \tag{60}$$

where the last line uses (38) and $\mathbb{E}\left[\boldsymbol{x}_b\boldsymbol{x}_b^\top \boldsymbol{A}\boldsymbol{x}_a\boldsymbol{x}_a^\top\right] = \begin{cases} \boldsymbol{\Lambda}\boldsymbol{A}\boldsymbol{\Lambda}, & a \neq b \\ 2\boldsymbol{\Lambda}\boldsymbol{A}\boldsymbol{\Lambda} + \mathrm{trace}(\boldsymbol{A}\boldsymbol{\Lambda})\boldsymbol{\Lambda}, & a = b \end{cases}$ in (Sayed, 2011, Lemma A.2).

For the second term of the last equation in (59), we can derive

$$\mathbb{E}[\boldsymbol{w}_{n+1}^\top\boldsymbol{x}_{n+1}\boldsymbol{x}_{n+1}^\top\boldsymbol{w}_{n+1}] = \mathbb{E}[\mathrm{trace}(\boldsymbol{\Lambda}\boldsymbol{w}_{n+1}\boldsymbol{w}_{n+1}^\top)]$$
$$= D_4\, \mathrm{trace}(\boldsymbol{\Lambda}), \tag{61}$$

where we define $D_4 = \begin{cases} \sigma_w^2 + (n+1)\sigma_e^2, & \gamma = 1 \\ \gamma^{2n+2}\sigma_w^2 + \frac{1-\gamma^{2n+2}}{1-\gamma^2}\sigma_e^2, & \gamma \neq 1 \end{cases}$ which follows the result in Lemma 1.

For the third term of the last equation in (59), we get

$$\mathbb{E}\left[\left(\sum_{i=1}^{n}\lambda^{n+1-i}\boldsymbol{w}_i^\top\boldsymbol{x}_i\boldsymbol{x}_i^\top\right)\widetilde{\boldsymbol{\Lambda}}^{-1}\boldsymbol{x}_{n+1}\boldsymbol{x}_{n+1}^\top\boldsymbol{w}_{n+1}\right] = \mathbb{E}\left[\,\mathrm{trace}\left(\sum_{i=1}^{n}\lambda^{n+1-i}\boldsymbol{\Lambda}\widetilde{\boldsymbol{\Lambda}}^{-1}\boldsymbol{\Lambda}\boldsymbol{w}_{n+1}\boldsymbol{w}_i^\top\right)\right]$$
$$= D_1\,\mathrm{trace}(\boldsymbol{\Lambda}\widetilde{\boldsymbol{\Lambda}}^{-1}\boldsymbol{\Lambda}), \qquad (62)$$

where the last line uses (31). This completes the proof.

$\square$

## D    PROOF OF THEOREM 3

*Proof.* We first expand $\mathbb{E}[(\widetilde{y}_{m+1} - \overline{y}_{m+1})^2]$ as

$$\mathbb{E}[(\widetilde{y}_{m+1} - \overline{y}_{m+1})^2]$$
$$= \mathbb{E}\left[\left(D_1\left(\sum_{i=1}^{m}\overline{\lambda}^{m+1-i}\overline{\boldsymbol{w}}_i^\top\overline{\boldsymbol{x}}_i\overline{\boldsymbol{x}}_i^\top\right)\widetilde{\boldsymbol{\Lambda}}^{-1}\overline{\boldsymbol{x}}_{m+1} - \overline{\boldsymbol{w}}_{m+1}^\top\overline{\boldsymbol{x}}_{m+1}\right)^2\right]$$
$$= D_1^2\,\mathbb{E}\left[\left(\sum_{i=1}^{m}\overline{\lambda}^{m+1-i}\overline{\boldsymbol{w}}_i^\top\overline{\boldsymbol{x}}_i\overline{\boldsymbol{x}}_i^\top\right)\widetilde{\boldsymbol{\Lambda}}^{-1}\overline{\boldsymbol{x}}_{m+1}\overline{\boldsymbol{x}}_{m+1}^\top\widetilde{\boldsymbol{\Lambda}}^{-1}\left(\sum_{j=1}^{m}\overline{\lambda}^{m+1-j}\overline{\boldsymbol{x}}_j\overline{\boldsymbol{x}}_j^\top\overline{\boldsymbol{w}}_j\right)\right]$$
$$+ \mathbb{E}[\overline{\boldsymbol{w}}_{m+1}^\top\overline{\boldsymbol{x}}_{m+1}\overline{\boldsymbol{x}}_{m+1}^\top\overline{\boldsymbol{w}}_{m+1}] - 2D_1\,\mathbb{E}\left[\left(\sum_{i=1}^{m}\overline{\lambda}^{m+1-i}\overline{\boldsymbol{w}}_i^\top\overline{\boldsymbol{x}}_i\overline{\boldsymbol{x}}_i^\top\right)\widetilde{\boldsymbol{\Lambda}}^{-1}\overline{\boldsymbol{x}}_{m+1}\overline{\boldsymbol{x}}_{m+1}^\top\overline{\boldsymbol{w}}_{m+1}\right].$$
$$(63)$$

For the first term of the last equation in (63), we have

$$\mathbb{E}\left[\left(\sum_{i=1}^{m}\overline{\lambda}^{m+1-i}\overline{\boldsymbol{w}}_i^\top\overline{\boldsymbol{x}}_i\overline{\boldsymbol{x}}_i^\top\right)\widetilde{\boldsymbol{\Lambda}}^{-1}\overline{\boldsymbol{x}}_{m+1}\overline{\boldsymbol{x}}_{m+1}^\top\widetilde{\boldsymbol{\Lambda}}^{-1}\left(\sum_{j=1}^{m}\overline{\lambda}^{m+1-j}\overline{\boldsymbol{x}}_j\overline{\boldsymbol{x}}_j^\top\overline{\boldsymbol{w}}_j\right)\right]$$
$$= \mathbb{E}\left[\,\mathrm{trace}\left(\sum_{i=1}^{m}\sum_{j=1}^{m}\overline{\lambda}^{2m+2-i-j}\overline{\boldsymbol{x}}_i\overline{\boldsymbol{x}}_i^\top\widetilde{\boldsymbol{\Lambda}}^{-1}\overline{\boldsymbol{\Lambda}}\widetilde{\boldsymbol{\Lambda}}^{-1}\overline{\boldsymbol{x}}_j\overline{\boldsymbol{x}}_j^\top\overline{\boldsymbol{w}}_j\overline{\boldsymbol{w}}_i^\top\right)\right]$$
$$= \overline{D}_2\,\mathrm{trace}(\overline{\boldsymbol{\Lambda}}\,\mathrm{trace}(\widetilde{\boldsymbol{\Lambda}}^{-1}\overline{\boldsymbol{\Lambda}}\widetilde{\boldsymbol{\Lambda}}^{-1}\overline{\boldsymbol{\Lambda}}) + 2\overline{\boldsymbol{\Lambda}}\widetilde{\boldsymbol{\Lambda}}^{-1}\overline{\boldsymbol{\Lambda}}\widetilde{\boldsymbol{\Lambda}}^{-1}\overline{\boldsymbol{\Lambda}}) + \overline{D}_3\,\mathrm{trace}(\overline{\boldsymbol{\Lambda}}\widetilde{\boldsymbol{\Lambda}}^{-1}\overline{\boldsymbol{\Lambda}}\widetilde{\boldsymbol{\Lambda}}^{-1}\overline{\boldsymbol{\Lambda}}), \quad (64)$$

where the last line uses (38) and $\mathbb{E}\left[\overline{\boldsymbol{x}}_b\overline{\boldsymbol{x}}_b^\top\boldsymbol{A}\overline{\boldsymbol{x}}_a\overline{\boldsymbol{x}}_a^\top\right] = \begin{cases}\overline{\boldsymbol{\Lambda}}\boldsymbol{A}\overline{\boldsymbol{\Lambda}}, & a \neq b \\ 2\overline{\boldsymbol{\Lambda}}\boldsymbol{A}\overline{\boldsymbol{\Lambda}} + \mathrm{trace}(\boldsymbol{A}\overline{\boldsymbol{\Lambda}})\overline{\boldsymbol{\Lambda}}, & a = b\end{cases}$ in (Sayed,

2011, Lemma A.2). By replacing $\lambda$, $\gamma$, $\sigma_w^2$, $\sigma_e^2$ and $n$ in $D_2$ and $D_3$ with $\overline{\lambda}$, $\overline{\gamma}$, $\overline{\sigma}_w^2$, $\overline{\sigma}_e^2$ and $m$, respectively, we obtain the modified quantities $\overline{D}_2$ and $\overline{D}_3$.

For the second term of the last equation in (63), we can derive

$$\mathbb{E}[\overline{\boldsymbol{w}}_{n+1}^\top\overline{\boldsymbol{x}}_{n+1}\overline{\boldsymbol{x}}_{n+1}^\top\overline{\boldsymbol{w}}_{n+1}] = \mathbb{E}[\mathrm{trace}(\overline{\boldsymbol{\Lambda}}\overline{\boldsymbol{w}}_{n+1}\overline{\boldsymbol{w}}_{n+1}^\top)]$$
$$= \overline{D}_4\,\mathrm{trace}(\overline{\boldsymbol{\Lambda}}) \qquad (65)$$

where we define $\overline{D}_4 = \begin{cases}\overline{\sigma}_w^2 + (m+1)\overline{\sigma}_e^2, & \overline{\gamma} = 1 \\ \overline{\gamma}^{2m+2}\overline{\sigma}_w^2 + \frac{1-\overline{\gamma}^{2m+2}}{1-\overline{\gamma}^2}\overline{\sigma}_e^2, & \overline{\gamma} \neq 1\end{cases}$ which follows the result in Lemma 1.

For the third term of the last equation in (63), we get

$$\mathbb{E}\left[\left(\sum_{i=1}^{m}\overline{\lambda}^{m+1-i}\overline{\boldsymbol{w}}_i^\top\overline{\boldsymbol{x}}_i\overline{\boldsymbol{x}}_i^\top\right)\widetilde{\boldsymbol{\Lambda}}^{-1}\overline{\boldsymbol{x}}_{m+1}\overline{\boldsymbol{x}}_{m+1}^\top\overline{\boldsymbol{w}}_{m+1}\right] = \mathbb{E}\left[\,\mathrm{trace}\left(\sum_{i=1}^{m}\overline{\lambda}^{m+1-i}\overline{\boldsymbol{\Lambda}}\widetilde{\boldsymbol{\Lambda}}^{-1}\overline{\boldsymbol{\Lambda}}\overline{\boldsymbol{w}}_{m+1}\overline{\boldsymbol{w}}_i^\top\right)\right]$$
$$= \overline{D}_1\,\mathrm{trace}(\overline{\boldsymbol{\Lambda}}\widetilde{\boldsymbol{\Lambda}}^{-1}\overline{\boldsymbol{\Lambda}}), \qquad (66)$$

where the last line uses (31). By replacing $\lambda$, $\gamma$, $\sigma_w^2$, $\sigma_e^2$ and $n$ in $D_1$ with $\overline{\lambda}$, $\overline{\gamma}$, $\overline{\sigma}_w^2$, $\overline{\sigma}_e^2$ and $m$, we obtain the modified quantity $\overline{D}_1$. This completes the proof.

$\square$

