# OpenReview forum: "Learning to Adapt: In-Context Learning Beyond Stationarity"
_ICLR.cc/2026/Conference — ICLR 2026 Poster_

### Official Review · Reviewer_V7fJ · 2025-10-22

**Soundness:** 3
**Presentation:** 3
**Contribution:** 3
**Rating:** 6
**Confidence:** 2

**Summary:**

This paper presents a theoretical analysis of in-context learning (ICL) for non-stationary regression tasks. By leveraging the linearity of both the gated in-context learner and the underlying task, the authors derive closed-form results for the convergence of weight parameters and provide quantitative expressions for training and testing errors. These errors are explicitly characterized in terms of the forgetting factor $\lambda$, the task dynamics parameter $\gamma$, and the token length $n$. The framework also unifies linear attention and gated linear attention through a single parameterization by $\lambda$, enabling a coherent theoretical comparison.

**Strengths:**

The paper leverages the linear structure of both the gated ICL model and underlying tasks to obtain the convergence results and explicit error bounds. The dependence of these bounds on $\lambda$, $\gamma$, $n$, provides the meaningful insights for practical model design and parameter selection. Overall, I think this paper makes a valuable contribution, providing the theoretical understanding of how gated linear transformers for non-stationary linear setting.

**Weaknesses:**

While this paper is theoretically strong and sufficiently novel, it is currently limited to linear models and tasks.
However, I think that these limitations are acceptable for the scope of this paper. Out of interest, I would like to ask the questions below.

**Questions:**

Do you expect the analysis can be extended to nonlinear settings, such as softmax attention or nonlinear task mappings ? If so, what kind of interpretation would be obtained. Would it be similar interpretation  to the linear case ?

---

> ### Author Response · Authors · 2025-11-21
> **Response to Reviewer V7fJ**
>
> Q1: While this paper is theoretically strong and sufficiently novel, it is currently limited to linear models and tasks. However, I think that these limitations are acceptable for the scope of this paper. Out of interest, I would like to ask the questions below.
>        Do you expect the analysis can be extended to nonlinear settings, such as softmax attention or nonlinear task mappings? If so, what kind of interpretation would be obtained. Would it be similar interpretation to the linear case?
>
>
>     A1:  Thank you for your insightful comment. First, convergence analysis of softmax attention under non-stationary environments is feasible, as discussed in [1] under stationary environments; however, deriving explicit expressions for training and testing errors is challenging, although upper bounds remain possible.
>
>     Second, we expect that nonlinear task mappings would yield interpretations similar to the linear case under non-stationary conditions. Direct theoretical analysis is challenging, which is why we complement our study with experiments on real-world datasets in the last two experiments. These datasets are typically nonlinear, and the results indicate that the GLA model can still identify an effective optimal λ.
>
>     Finally, we have added a future work direction to generalize the first-order autoregressive assumption to a broader class of dynamic-weight models. In particular, considering more flexible temporal evolutions—such as higher-order dynamics, stochastic drift, or slowly varying adversarial changes-would further illuminate how in-context learning behaves in general non-stationary settings.
>
>     [1] Yu Huang, Yuan Cheng, and Yingbin Liang. In-context convergence of transformers. Forty-first International Conference on Machine Learning, 2024.

---

> > ### Comment · Reviewer_V7fJ · 2025-11-26
> >
> > Thank you very much for the clarification. I would like to keep my score.

---

> > > ### Author Response · Authors · 2025-11-26
> > >
> > > Thank you for your response and acknowledgment of our rebuttal. We are glad that our rebuttal addressed concerns and we would be happy to address any further questions or suggestions you may have.

---

### Official Review · Reviewer_xcya · 2025-10-24

**Soundness:** 2
**Presentation:** 3
**Contribution:** 2
**Rating:** 4
**Confidence:** 3

**Summary:**

This paper provides a theoretical analysis of in-context learning (ICL) under non-stationary regression problems, where task distributions evolve over time. It demonstrates that Gated Linear Attention (GLA) outperforms standard linear attention by adaptively modulating past inputs through a learnable recency bias, enabling better handling of time-varying functions. Theoretical results and empirical validation confirm GLA's advantages in dynamic settings.

**Strengths:**

1. The paper introduces a novel theoretical framework for in-context learning (ICL) under non-stationary regression problems.
2.  The work exhibits certain technical rigor, with precise theoretical derivations.
3. The paper is well-structured, clearly defining the non-stationary ICL problem early and motivating it with real-world examples (e.g., time-series forecasting).

**Weaknesses:**

1. The theoretical analysis focuses exclusively on linear regression tasks with Gaussian inputs, which is a very narrow setting. Real-world in-context learning scenarios often involve non-linear functions, complex data distributions, and high-dimensional inputs.
2. The paper models non-stationarity using a first-order autoregressive process, which is a highly constrained form of time-varying dynamics. This fails to capture more complex real-world non-stationarity patterns such as abrupt changes, periodic variations, or long-term trends.
3. The paper briefly shows that deeper GLA models perform better but provides no theoretical justification or detailed analysis of how gating mechanisms interact across multiple layers in non-stationary settings.
4. While the paper includes a sentiment classification experiment on SST-2, this is only a single NLP task and doesn't demonstrate the broader applicability of GLA to diverse non-stationary scenarios.
5. While the paper briefly mentions adaptive signal processing methods in Section 3, it doesn't actually compare GLA's performance with established methods like RLS or LMS in the same non-stationary regression setting

**Questions:**

See the Weaknesses.

---

> ### Author Response · Authors · 2025-11-21
> **Response to Reviewer xcya**
>
> Q1: The theoretical analysis focuses exclusively on linear regression tasks with Gaussian inputs, which is a very narrow setting. Real-world in-context learning scenarios often involve non-linear functions, complex data distributions, and high-dimensional inputs.
>
>     A1: Thank you for raising this point. Our focus on Gaussian inputs and linear regression tasks follows a well-established line of work in the theoretical study of in-context learning [1–8]. Gaussian features permit sharp, explicit characterizations of both training and test errors, avoiding the need for loose worst-case bounds, while linear regression provides a clean setting in which the dynamical behavior and error propagation of in-context learning algorithms can be precisely analyzed. This analytical tractability is essential for revealing how key quantities such as γ and λ govern the behavior of the learned in-context learner.
>
>     While Gaussian inputs represent a stylized setting, they provide a mathematically clean foundation that has proven indispensable for theoretical progress in this area. Importantly, our goal in this work is not to model all real-world input distributions but to understand, at a fundamental level, how gated linear attention can learn dynamic regression tasks in a non-stationary environment. This requires a setting where exact solutions and convergence dynamics can be derived, as in prior studies.
>
>     We also note that our empirical results go beyond the Gaussian assumption: the real-world datasets used in our experiments feature non-Gaussian, non-linear, and higher-dimensional structures. These experiments demonstrate that the insights derived under the Gaussian setting extend cleanly to more complex data distributions.
>
>     We have revised the manuscript in line 169 to clarify the motivation for this modeling choice and its consistency with the theoretical literature.
>
>     [1] Kwangjun Ahn, Xiang Cheng, Hadi Daneshmand, and Suvrit Sra. Transformers learn to implement preconditioned gradient descent for in-context learning. Advances in Neural Information Processing Systems,  2023.
>     [2] Yu Huang, Yuan Cheng, and Yingbin Liang. In-context convergence of transformers. Forty-first International Conference on Machine Learning, 2024.
>     [3] Siyu Chen, Heejune Sheen, Tianhao Wang, and Zhuoran Yang. Training dynamics of multi-head softmax attention for in-context learning: Emergence, convergence, and optimality. The Thirty Seventh Annual Conference on Learning Theory, PMLR, 2024.
>     [4] Tong Yang, Yu Huang, Yingbin Liang, and Yuejie Chi. In-context learning with representations: Contextual generalization of trained transformers. Advances in Neural Information Processing Systems, 2024.
>     [5] Yingcong Li, Ankit S Rawat, and Samet Oymak. Fine-grained analysis of in-context linear estimation: Data, architecture, and beyond. Advances in Neural Information Processing Systems, 2024
>     [6] Arvind Mahankali, Tatsunori B Hashimoto, and Tengyu Ma. One step of gradient descent is provably the optimal in-context learner with one layer of linear self-attention. The Twelfth International Conference on Learning Representations, 2024.
>     [7] Ruiqi Zhang, Spencer Frei, and Peter L Bartlett. Trained transformers learn linear models in-context. Journal of Machine Learning Research, 2024.
>     [8] Yedi Zhang, Aaditya K Singh, Peter E Latham, and Andrew Saxe. Training dynamics of in-context learning in linear attention. Forty-second International Conference on Machine Learning, 2025.

---

> ### Author Response · Authors · 2025-11-21
> **Response to Reviewer xcya**
>
> Q2: The paper models non-stationarity using a first-order autoregressive process, which is a highly constrained form of time-varying dynamics. This fails to capture more complex real-world non-stationarity patterns such as abrupt changes, periodic variations, or long-term trends.
>
>     A2: Thank you for pointing this out. As we also noted in our response to Reviewer WfeF, prior theoretical studies on in-context learning [1–8] focus exclusively on regression problems with stationary target functions. In contrast, our work provides, to the best of our knowledge, the first theoretical analysis of in-context learning under non-stationary or time-varying regression problems.
>
>     We agree that the first-order autoregressive model in Eq.(3) represents a simple form of non-stationarity. However, this model is sufficiently expressive to clearly illustrate the convergence behavior, the training dynamics of gated linear attention, and the generalization performance of the resulting in-context learner. More importantly, to assess robustness beyond this simplified model, we additionally conduct experiments on real-world datasets whose underlying mappings are non-stationary and do not follow a first-order autoregressive pattern. These experiments provide broader empirical evidence that complements our theoretical findings.
>
>     In the revised manuscript, we have also expanded our discussion of future directions, clarifying how our framework could be extended to richer classes of dynamic-weight models. In particular, we outline possible generalizations to higher-order temporal dynamics, stochastic drift, or slowly varying adversarial changes, which would support a more comprehensive theoretical treatment of in-context learning in general non-stationary environments.
>
> Q3: The paper briefly shows that deeper GLA models perform better but provides no theoretical justification or detailed analysis of how gating mechanisms interact across multiple layers in non-stationary settings.
>
>     A3: Thank you for your insightful comment.  Conceptually, each GLA layer implements a linear adaptive filter whose effective behavior is determined by its weights. When multiple layers are stacked, these adaptive filters operate at different timescales, enabling the network to simultaneously capture short-term fluctuations and longer-term trends in the evolving regression weights. This multi-timescale structure explains why deeper GLA models achieve better performance under non-stationary regression: a single layer can track only one effective timescale of drift, while multiple layers collectively approximate a richer family of dynamic predictors. A full theoretical analysis of how multiple GLA layers interact and form a multi-timescale adaptive structure is beyond the scope of the current work, and we will address this direction in future work.
>
>     We have added this conceptual discussion to second experiment and  noted the theoretical extension in the conclusion.
>
> Q4: While the paper includes a sentiment classification experiment on SST-2, this is only a single NLP task and doesn't demonstrate the broader applicability of GLA to diverse non-stationary scenarios.
>
>     A4: Thank you for your comment. To further demonstrate the broader applicability of GLA to diverse and non-stationary NLP scenarios, we additionally evaluate its ICL capability on a more challenging natural language inference task. Specifically, we use the Multi-Genre Natural Language Inference (MNLI) dataset, which spans a wide range of text genres and contains approximately 393k training examples with three labels (entailment, contradiction, and neutral). Following the same setup as in the third experiment, we provide 10 in-context demonstrations per instance for ICL fine-tuning—constrained by context length—and compute loss only on the label tokens. During evaluation, we vary the number of demonstrations from {1, 3, 5, 7, 10}. As shown in the last experiment, GLA consistently achieves higher accuracy and confidence than LA, highlighting the advantage of the gating mechanism on a multi-genre, non-stationary task.

---

> ### Author Response · Authors · 2025-11-21
> **Response to Reviewer xcya**
>
> Q5: While the paper briefly mentions adaptive signal processing methods in Section 3, it doesn't actually compare GLA's performance with established methods like RLS or LMS in the same non-stationary regression setting.
>
>     A5: Thank you for your suggestion. In the previous version, we did not include a comparison between the GLA model and LMS/RLS because our focus was on examining the theoretical properties of GLA in the context of in-context learning. Following the reviewer’s recommendation, we have now incorporated the corresponding simulation results in the first experiment. As mentioned previously a one-layer GLA model applied to a first-order autoregressive process functions analogously to an adaptive filter. To illustrate this, we compare its performance with LMS and RLS algorithms. We set the LMS step size to 0.01 and the RLS forgetting factor to 0.98, train on sequences of length 1000, and perform 10,000 Monte Carlo trials, averaging the results. The training errors for LMS and RLS are respectively [0.2639  0.3168   0.6058  1.0072  1.4758 ] and [0.2555 0.3746  0.6658  0.8881  1.2916] for γ = [0.8  0.85  0.925  0.95  0.975]. Compared to LMS and RLS, which require fixed or slowly adapting parameters, a one-layer GLA model achieves lower training errors (see Figure 1 (a)) because it possesses higher representational flexibility. Furthermore, LMS and RLS adapt only to a single sequence at a time, requiring retraining for each new input, and therefore cannot leverage cross-sequence information. In contrast, GLA's learnable weights are shared across sequences, allowing the model to generalize and adapt efficiently to new inputs without retraining.

---

> > ### Comment · Reviewer_xcya · 2025-11-24
> >
> > Thanks for your response. It has addressed my main concerns, and I will improve the rating from 4 to 6.

---

> > > ### Author Response · Authors · 2025-11-25
> > >
> > > Thank you for your response and acknowledgment of our rebuttal. We are glad that our rebuttal addressed concerns and we would be happy to address any further questions or suggestions you may have.

---

### Official Review · Reviewer_WfeF · 2025-10-25

**Soundness:** 3
**Presentation:** 3
**Contribution:** 3
**Rating:** 6
**Confidence:** 3

**Summary:**

In this paper, the authors study the capability of ICL to learn time-varying function class, thus provides a valuable addition to literature. In particular, the authors provide theoretical results on the convergence of the gradient flows, as well as training and testing errors. Numerical experiments are also provided for validation.

**Strengths:**

The authors provide rigorous theoretical research on the congruence of the gradient flows for ICL training, and quantify the training and testing errors. The analysis is correct and detailed.

**Weaknesses:**

1. The function class considered is limited. The analysis is restricted to the case in equation (3), it is desired to demonstrate the implications of the results for more broad classes of functions.
2. The numerical experiments are limited. Given strong assumptions made on initialization for the theoretically results, numerical experiments offer a good opportunity to explore what happens when those assumptions are violated.

**Questions:**

1. Can you provide comments on the behavior of the gradient flow when the initialization assumptions in both Assumption 1 and the Theorem 1 do not hold?
On Lines 146-147, e_i and \eta_i are not consistent.

---

> ### Author Response · Authors · 2025-11-21
> **Response to Reviewer WfeF**
>
> Q1: The function class considered is limited. The analysis is restricted to the case in equation (3), it is desired to demonstrate the implications of the results for more broad classes of functions.
>
>     A1:  Thank you for pointing this out. Prior theoretical studies on in-context learning [1–8] focus exclusively on regression problems with stationary target functions. In contrast, our work provides, to the best of our knowledge, the first theoretical analysis of in-context learning under non-stationary or time-varying regression problems.
>
>     We agree that the first-order autoregressive model in Eq. (3) is a simple instance of non-stationarity. However, this model is sufficiently expressive to clearly demonstrate the convergence behavior, training dynamics of gated linear attention, and the generalization performance of the resulting in-context learner. More importantly, to evaluate the implications beyond this simple model class, we additionally conduct experiments on real-world datasets in the last two experiments, whose underlying mappings are non-stationary and do not follow a first-order autoregressive structure. These experiments provide broader empirical evidence that complements our theoretical findings.
>
>     In the revised manuscript, we have also expanded the discussion of future work to clarify how our framework can be extended to a broader range of dynamic-weight models. In particular, we highlight the possibility of analyzing more general temporal evolutions of the regression weights--such as higher-order dynamics, stochastic drift, or slowly varying adversarial changes--which would allow a more comprehensive theoretical understanding of in-context learning under general non-stationary environments.
>
>     [1] Kwangjun Ahn, Xiang Cheng, Hadi Daneshmand, and Suvrit Sra. Transformers learn to implement preconditioned gradient descent for in-context learning. Advances in Neural Information Processing Systems,  2023.
>     [2] Yu Huang, Yuan Cheng, and Yingbin Liang. In-context convergence of transformers. Forty-first International Conference on Machine Learning, 2024.
>     [3] Siyu Chen, Heejune Sheen, Tianhao Wang, and Zhuoran Yang. Training dynamics of multi-head softmax attention for in-context learning: Emergence, convergence, and optimality. The Thirty Seventh Annual Conference on Learning Theory, PMLR, 2024.
>     [4] Tong Yang, Yu Huang, Yingbin Liang, and Yuejie Chi. In-context learning with representations: Contextual generalization of trained transformers. Advances in Neural Information Processing Systems, 2024.
>     [5] Yingcong Li, Ankit S Rawat, and Samet Oymak. Fine-grained analysis of in-context linear estimation: Data, architecture, and beyond. Advances in Neural Information Processing Systems, 2024
>     [6] Arvind Mahankali, Tatsunori B Hashimoto, and Tengyu Ma. One step of gradient descent is provably the optimal in-context learner with one layer of linear self-attention. The Twelfth International Conference on Learning Representations, 2024.
>     [7] Ruiqi Zhang, Spencer Frei, and Peter L Bartlett. Trained transformers learn linear models in-context. Journal of Machine Learning Research, 2024.
>     [8] Yedi Zhang, Aaditya K Singh, Peter E Latham, and Andrew Saxe. Training dynamics of in-context learning in linear attention. Forty-second International Conference on Machine Learning, 2025.
>
> Q2: The numerical experiments are limited. Given strong assumptions made on initialization for the theoretically results, numerical experiments offer a good opportunity to explore what happens when those assumptions are violated. Can you provide comments on the behavior of the gradient flow when the initialization assumptions in both Assumption 1 and the Theorem 1 do not hold?
>
>     A2: Thank you for your comment. In fact, our numerical experiments already use a standard random Gaussian initialization rather than the structured initialization required by Assumption 1 and Theorem 1, and we realized that this was not clearly highlighted in the paper. Despite violating the theoretical initialization constraints, the gradient flow dynamics in our experiments still match the predicted behavior, suggesting that the constraint is not essential in practice.
>
>     From a theoretical standpoint, the model admits a global optimization, and random initialization is sufficient to reach a global minimum under gradient flow, as also reflected in our empirical results. A more complete theoretical characterization of gradient flow without the initialization assumptions is an important direction that we will pursue in future work, and we have added this discussion to the conclusion.
>
> Q3: On Lines 146-147, $e_i$ and $\eta_i$ are not consistent.
>
>     A3: Thank you for your comment.  We have corrected it in the revised manuscript.

---

> > ### Comment · Reviewer_WfeF · 2025-11-26
> >
> > Thank you very much for the clarification. I would like to maintain my score.

---

> > > ### Author Response · Authors · 2025-11-26
> > >
> > > Thank you for your response and acknowledgment of our rebuttal. We are glad that our rebuttal addressed concerns and we would be happy to address any further questions or suggestions you may have.

---

### Author Response · Authors · 2025-12-02
**Rebuttal Summary**

We sincerely regret the recent incident and appreciate the extra effort it may have required from the AC. We would like to thank the reviewers for their constructive feedback on our work. Below, we summarize our responses to the discussion.

The main topic raised concerned the theoretical understanding of in-context learning in nonstationary environments. In the revised version, we have addressed all questions raised by the three reviewers. Further details are provided in the individual rebuttals.

Reviewers WfeF and V7fJ (Nov. 26) maintained their score of 6, supporting acceptance. Reviewer xcya (Nov. 23) raised the score from 4 to 6, noting that the concerns had been addressed.

Although we believe some of the points raised did not fundamentally challenge the core contributions of our work, overall we found the discussion constructive and thoughtful. It has helped us further refine the manuscript through new experiments and various improvements.

We sincerely thank the reviewers and the AC for their time and engagement throughout the review and rebuttal process.

---

### Meta-Review · Area_Chair_2xYa · 2026-01-03

**Summary:**

The paper studies in-context learning under non-stationary regression, with a focus on how gated linear attention adapts when the target evolves over time. Its main contribution is a theoretical characterization of training and test errors and learning dynamics showing that GLA implicitly learns a recency bias and can outperform standard linear attention in drifting environments, supported by controlled simulations and additional experiments. The reviewers raised several concerns, but these were addressed to a degree that makes the contribution clear and coherent. First, while the theory is developed for a restricted and stylized setting (linear regression with Gaussian inputs and an AR(1) drift), the authors convincingly position this as a minimal yet expressive model that enables sharp analysis, and they complement it with real-world non-stationary experiments that demonstrate the qualitative relevance of the findings beyond the formal assumptions. Second, concerns about strong initialization assumptions were effectively mitigated by clarifying that the reported experiments already violate these assumptions through standard random initialization while still matching theoretical predictions, which strengthens confidence in the robustness of the conclusions. Third, the rebuttal broadened the empirical and conceptual scope by adding comparisons to classical adaptive filtering methods such as LMS and RLS, directly addressing the signal-processing angle and highlighting how GLA differs by sharing parameters across sequences rather than adapting per instance. Finally, although extensions to richer non-stationarity patterns and nonlinear or softmax-based attention remain largely theoretical and are discussed rather than proven, the authors are transparent about these limitations and frame them appropriately as future directions, supported by empirical evidence suggesting similar behavior. Overall, the paper provides a solid and timely theoretical contribution to understanding in-context learning under drift, the experimental support is adequate for the scope of the theory, and the reviewers’ concerns have been addressed sufficiently to warrant acceptance.

**Reviewer Concerns:**

The main reviewer concerns centered on the narrowness of the theoretical setting, the realism of the non-stationarity model, the strength of the initialization assumptions, and the limited empirical scope. The rebuttal addressed most of these points convincingly. The restriction to linear regression with Gaussian inputs and AR(1) drift remains a limitation, but it was adequately justified as a tractable core model, and the added real-world experiments helped alleviate worries about relevance beyond this stylized case. Concerns about initialization assumptions were effectively resolved by clarifying that the experiments already use standard random initialization and still follow the predicted dynamics. Requests for broader empirical validation were partially addressed through additional NLP experiments and explicit comparisons with LMS and RLS, which strengthened the paper’s positioning relative to adaptive filtering. What remains outstanding is a fully developed theory for deeper or nonlinear attention models and for more complex forms of non-stationarity; however, these gaps are clearly acknowledged and reasonably deferred to future work rather than undermining the current contribution.

**Reviewer Scores:**

- WfeF (initial score: 6): Likely no change. The reviewer’s main questions on function class limitations and initialization were directly answered, and they explicitly stated satisfaction and maintained their score.
- xcya (initial score: 4): Increased to 6. The reviewer initially raised multiple substantive concerns about modeling assumptions, empirical breadth, and comparisons, all of which were addressed through clarifications, added experiments, and new comparisons, leading them to raise their score during the discussion.
- V7fJ (initial score: 6): Likely no change. This reviewer already viewed the paper positively, and the discussion around nonlinear extensions and interpretation clarified open questions without fundamentally altering their assessment.

---

### Decision · Program_Chairs · 2026-01-26

Accept (Poster)